# The role of arsenic in the operation of sulfur-based electrical threshold switches

Renjie Wu[1,2,8], Rongchuan Gu[3,8], Tamihiro Gotoh [4,8], Zihao Zhao[1,2], Yuting Sun[1,2], Shujing Jia[5], Xiangshui Miao [3], Stephen R. Elliott[6,7], Min Zhu [1] ✉, Ming Xu [3] ✉ & Zhitang Song [1] ✉

Arsenic is an essential dopant in conventional silicon-based semiconductors and emerging phase-change memory (PCM), yet the detailed functional mechanism is still lacking in the latter. Here, we fabricate chalcogenide-based ovonic threshold switching (OTS) selectors, which are key units for suppressing sneak currents in 3D PCM arrays, with various As concentrations. We discovered that incorporation of As into GeS brings >100 °C increase in crystallization temperature, remarkably improving the switching repeatability and prolonging the device lifetime. These benefits arise from strengthened As-S bonds and sluggish atomic migration after As incorporation, which reduces the leakage current by more than an order of magnitude and significantly suppresses the operational voltage drift, ultimately enabling a back-end-of-line-compatible OTS selector with >12 MA/cm² on-current, ~10 ns speed, and a lifetime approaching $10^{10}$ cycles after 450 °C annealing. These findings allow the precise performance control of GeSAs-based OTS materials for high-density 3D PCM applications.

Arsenic (As) is a hypertoxic element, yet it has long been extensively used in semiconductor manufacturing[1]; e.g., the n-type semiconductors fabricated by ion implantation of As into silicon substrates are the building blocks of modern transistors[2]. Meanwhile, As is also a principal component of gallium arsenide, the landmark material of second-generation semiconductors[3]. Moreover, As has never been absent in the discovery, development, and eventual commercialization of phase-change memory (PCM)[4–6], an emerging memory technology to bridge the large performance gap between Flash and DRAM in modern computers[7,8]. Electrical-switching behavior[9] was discovered as early as 1964 in As-based chalcogenides, i.e., As-Te-I, by Northover and Pearson[10], as well as in the As-Te-Se system by Dennard in 1966[11]. Two years later, Ovshinsky reported the repeatable ovonic threshold switching (OTS) phenomenon in the amorphous state of

$As_{30}Te_{48}Si_{12}Ge_{10}$[12], that is, the material becomes highly conductive abruptly once the voltage bias reaches a threshold value and it returns to the low-conductance state when the voltage is removed. Interestingly, if the concentration of As is reduced below 5 at.%, the large current at the threshold-switching point could heat up and crystallize the material, turning it into a permanent low-resistance state. One only needs to melt and quench the crystal to obtain the high-resistance amorphous state again; this is the working principle of PCM.

The OTS device, however, has no memory effect, and nowadays is usually used as a key selector component in 3D PCM integration. In traditional PCM chips, each memory unit is connected with a transistor to control the opening and shutting of this unit, while in 3D PCM chips, a three-terminal transistor is too bulky to fit into the compact structure, and thus an OTS selector, due to its easy fabrication and high

[1]National Key Laboratory of Materials for Integrated Circuits, Shanghai Institute of Microsystem and Information Technology, Chinese Academy of Sciences, Shanghai 200050, China. [2]University of Chinese Academy of Sciences, Beijing 100029, China. [3]Wuhan National Laboratory for Optoelectronics, School of Integrated Circuits, Huazhong University of Science and Technology, Wuhan 430074, China. [4]Department of Physics, Graduate School of Science and Technology, Gunma University, Maebashi 3718510, Japan. [5]Frontier Institute of Chip and System, Fudan University, Shanghai 200050, China. [6]Trinity College, University of Cambridge, Cambridge CB2 1TQ, UK. [7]Physical and Theoretical Chemistry Laboratory, University of Oxford, Oxford OX1 3QZ, UK. [8]These authors contributed equally: Renjie Wu, Rongchuan Gu, Tamihiro Gotoh. ✉e-mail: minzhu@mail.sim.ac.cn; mxu@hust.edu.cn; ztsong@mail.sim.ac.cn

compatibility with PCM, becomes the best replacement. Yet, the performance of OTS selectors is not as satisfactory as traditional transistors, particularly in terms of their stability and on/off ratios; for example, most OTS materials can hardly withstand the back-end-of-line (BEOL) processing temperature (400 °C–450 °C)[8]. Thus, searching for OTS materials with high crystallization temperatures, and without compromising their other performances, is the key task to fully commercialize 3D PCM devices. To date, a lot of materials have been discovered that exhibit OTS behavior, including As-free materials, such as Se-Te[13], Zn-Se[13], GeTe6[14], Ge-Se[15], Si-Te[16], GeS[17] etc., but an As-based OTS material (As-Se-Ge-Si)[18] is the only one that has been successfully used in actual 3D PCM chips (e.g., two decks, 128 Gb, by Intel in 2017)[19]. Thanks to the vertical stacking ability of OTS selector on the PCM layer, unlike silicon-based selectors that only survive on silicon substrates, a four-deck stacked PCM with a 256 GB device was recently released[18], comparable with advanced 3D Flash memory.

The function of As in traditional *n*-type semiconductors (e.g., crystalline Si) has been well studied, that is, providing extra free electrons for conduction via the substitution of Si (four outer electrons) by As atom dopants (five outer electrons), while the role that As plays in OTS materials is still the subject of heated debate. Cheng et al. attributed the increase in the thermal stability of Ge-As-Se to the presence of As[20]. Garbin et al. showed that the incorporation of As made a difference in inhibiting elemental segregation from elemental mappings, thus extending the device lifetime[21]. Noé et al. believed that As would help prevent oxidation, thereby ameliorating the endurance or thermal stability of the device[22]. From the perspective of electron energy bands, Adriaenssens[23] argued that the introduction of As would bring new defect states into the bandgap. Although reports of previous research have partly mentioned the function of As, precisely how it works in OTS materials and the mechanism behind it is still lacking, which has remarkably slowed down the research and development of next-generation 3D PCM devices.

## Results

To reveal the role of As in the performance of OTS selectors, we employed GeS as a prototype material, for which there are already clear device performance parameters and energy-band data[17,24]. We added 0, 20, 25, and 43 at.% As into GeS, abbreviated as GeS, GeSAs20, GeSAs25, and GeSAs43, respectively. T-shaped devices for each composition were fabricated, and transmission electron microscopy (TEM) photos of them are shown in Fig. 1a, in which we deposited 40 nm TiN as the top electrode, and the 5 nm thick carbon layer below serves as a buffer layer to prevent elemental diffusion. The bottom electrode is a cylindrical TiN electrode with a diameter of 200 nm. As displayed in energy-dispersive spectroscopy (EDS) mappings, Ti, C, Ge, As, and S elements are homogeneously distributed without segregation or diffusion even after multiple operations. Figure 1b displays DC current-voltage ($I$–$V$) curves of GeSAs devices with different As contents, in which a series-connected 3.3 kΩ resistor was embedded to avoid surge currents. Before regular threshold-switching operation, a 3.8 V fire voltage ($V_{fire}$) initializes the GeSAs devices, as described by the dashed lines. Then, the switching voltage denoted by the threshold voltage ($V_{th}$) sharply decreases to -1.5 V for pure GeS devices, as described by the solid lines. Interestingly, GeSAs20 cells exhibited a ~3.0 V $V_{th}$ increased by 1.5 V, which further drop to -2.4 V and to -2 V for GeSAs25 and GeSAs43 cells, respectively. Obviously, the incorporation of 20 at.% As increases $V_{th}$ the most but then it decreases as one continues to add more As. Meanwhile, the leakage current at $1/2V_{th}$, known as $I_{off}$, is the smallest in GeSAs20 devices. $I_{off}$ is only ~15 nA in this device, almost a factor of ten better than that in pure GeS devices (-140 nA). However, continuing to increase the As concentration fails to further reduce the leakage current, e.g., $I_{off}$ increases to 40 nA for GeSAs43 devices.

As also plays an important role in the uniformity and endurance of selector devices, as shown in the current-voltage ($I$–$V$) curves of GeSAs devices under 100 continuous triangular pulses (Fig. 1c), and the performance statistics of individual devices (Fig. S1). All the devices were successfully switched on at -2 V and turned off at -1.5 V (the holding

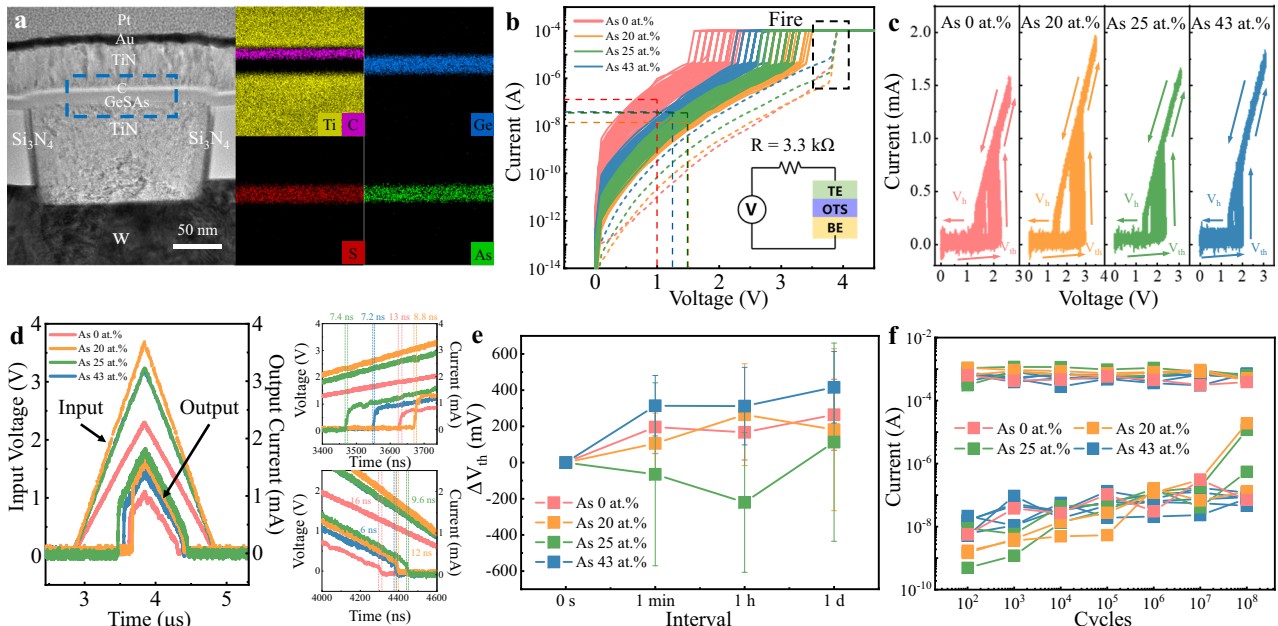

**Fig. 1 | Structure and electrical characteristics of GeSAs devices. a** Cross-sectional transmission electron microscopy (TEM) image of a GeSAs25 device, subjected to several triangular pulses, and energy-dispersive spectroscopy (EDS) mappings of Ti, C, Ge, S, and As. **b** DC $I$–$V$ curves of devices with various As contents. The dashed lines refer to the first-fire (FF) process, and the solid lines refer to $I_{off}$. The inset shows a schematic of the DC testing circuit. **c** $I$–$V$ curves of GeSAs devices subjected to 100 consecutive triangular pulses. **d** Input voltage and output current are integrated in the left panel, and the separate pictures are shown in the right panel. The operating speed of each component reaches the nanosecond level. **e** Drift of $V_{th}$ with time. Particular As contents can effectively diminish the spontaneous drift of $V_{th}$. **f** Endurance performance in the as-deposited state for all compositions. The $I_{on}$ is very stable, whereas the $I_{off}$ fluctuates with an apparent upward drift. We assume that the increase in $I_{off}$ may be jointly caused by phase separation or the formation of Ge-Ge filaments.

voltage, $V_h$). $V_{th}$ of pure GeS devices fluctuates within a range of 0.7 V (1.6 V–2.3 V), while the range is just 0.4 V (1.6 V–2 V) for GeSAs$_{43}$ cells, indicating that the addition of As can indeed improve the switching repeatability, whereas it hardly influences $V_h$ for all devices. Nevertheless, it does affect the on-current ($I_{on}$) captured at the point of threshold switching. It is known that the RESET switching of PCMs needs sufficient energy to melt the memory materials, thus requiring a large $I_{on}$ to be provided by the selectors. The average value of $I_{on}$ goes up from 0.59 mA for GeS devices to 1.36 mA for GeSAs$_{20}$ devices and then slightly down to 1.11 mA for the devices with 43 at.% As concentration (Fig. S2). The value of $I_{on}$ for GeS in this work is smaller than the DC results of a previous study[17], due to the series resistance (1.1 kΩ) employed and the diffusion of Al top electrode in previous work, but is consistent with the *I-V* curves from subsequent work with the same structure[24]. Since $I_{on}$ is almost size-independent, the current density of GeSAs devices sharply increases to >20 MA/cm² as the device size scales down to 60 nm, higher than that of Ge-Se/Te-based OTSs (Fig. S3). Although $V_{th}$ is closely determined by the As content, the switching speed seems to be As-independent, as shown in Fig. 1d and Fig. S4. The switching-speed test circuit is shown in Fig. S4a. We carried out speed tests on 30 different devices for each material composition and obtained the results in Fig. S4d statistically. The on-speeds of all compositions lie between 7 and 12 ns, and off-speeds are between 5 and 15 ns. A composition-independent switching speed is predominantly due to the electronic nature of the OTS behavior, in which atomic migration is barely involved[25,26].

Other benefits of As incorporation can be found in the $V_{th}$ drift of devices with time (Fig. 1e), that is, $V_{th}$ spontaneously increases over time after the first-fire (FF) process. The $V_{th}$ drift could induce write/read failure in the high-density memory array as well as the degradation of the device lifetime[27]. For pure GeS, the average $V_{th}$ increases by 196 mV within 1 min and then further rises up another 68 mV after

1 day. In the same way, the variation value of $V_{th}$ of GeSAs$_{20}$ starts from 106 mV, then increases to 265 mV, and stops at 182 mV. The rising trend of $V_{th}$ was inhibited at 25 at.% As content, where the variation value of $V_{th}$ decreases by 65 mV @ 1 min and eventually goes up 113 mV @ 1 day, while the increase in $V_{th}$ climbs up from 314 mV to 416 mV @ 1 day in GeSAs$_{43}$. Obviously, an appropriate As content is sufficient to restrain the $V_{th}$ drift. For device endurance, all GeS and GeSAs devices can be successfully turned on and off for at least 10⁸ cycles, as shown in Figs. S5, S6, and Fig. 1f.

OTS selectors must withstand a temperature of 450 °C for 30 minutes in the BEOL process, in which the metal wire is bonded and the insulator layer is deposited[28,29]. Since only OTS selectors in the amorphous state exhibit threshold-switching behavior[8], if they transform into crystals under this condition, the crystallized selectors will lose the OTS function and can no longer be recovered. We, therefore, studied the crystallization temperature of GeSAs films utilizing X-ray diffraction (XRD) in Fig. S7. After annealing at different temperatures, a crystalline peak emerges in the GeS diffractogram after undergoing a heat treatment at 400 °C for 10 minutes, whereas other three films containing As remained amorphous through annealing at 450 °C for 30 minutes and even at 500 °C for 10 minutes. That is to say, incorporation of 20 at.% As brings a more than 100 °C increase in the crystallization temperature, directly indicating that the incorporation of As contributes to a strong reinforcement of thermal stability, which is also identified by the Raman results of annealed GeSAs films (Fig. S8). The material morphology can be further confirmed by TEM and the corresponding fast Fourier transform (FFT) images of a GeSAs$_{43}$ device annealed at 450 °C for 30 minutes are shown in Fig. 2a. In addition, there was no observed segregation and diffusion of elements after high-temperature treatment on the basis of EDS mappings. We then applied 100 triangular electrical pulses to the annealed cells, and the corresponding *I-V* results are shown in Fig. 2b. After 450 °C annealing,

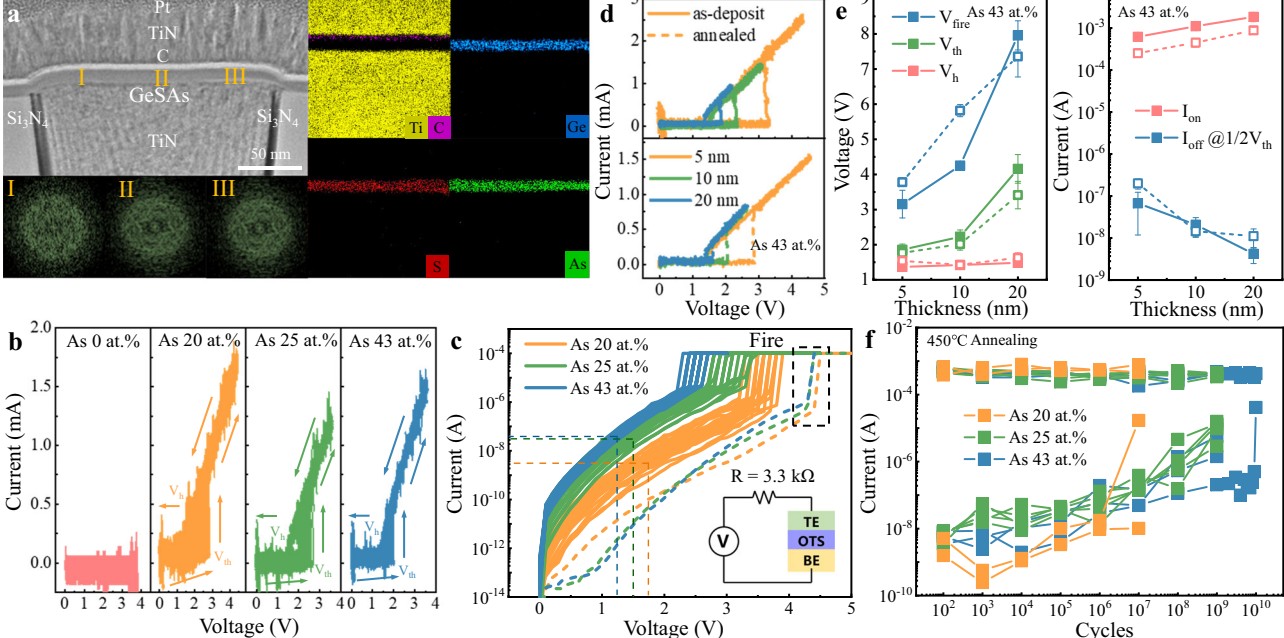

**Fig. 2 | Microstructure and electrical performance of GeSAs devices with different thicknesses after 450 °C annealing. a** TEM picture of a GeSAs$_{43}$ device after 450 °C annealing. FFT images labeled I, II, and III represent left, middle, and right areas marked on the GeSAs layer, revealing an amorphous nature. EDS mappings after heat treatment reveal no difference from those for the as-deposited state. **b** *I–V* curves of 450 °C annealed devices subject to 100 consecutive triangular pulses. 100 consecutive triangular pulses with the amplitudes of 4, 4.5, 4, and 4 V are applied to GeS, GeSAs$_{20}$, GeSAs$_{25}$, and GeSAs$_{43}$ annealed devices, respectively. The GeS device fails, whereas the GeSAs ones continue to work normally. **c** DC *I–V* curves of 450 °C annealed devices. **d** Different responses of GeSAs$_{43}$ devices with different thicknesses of the OTS layer before and after annealing. **e** Electrical performance for as-deposited and annealed GeSAs$_{43}$ devices with different thicknesses. The solid and dashed lines represent the device performance before and after annealing, respectively. $V_{fire}$, $V_{th}$, $V_h$, and $I_{on}$ of both states increase while $I_{off}$ monotonically decreases. **f** Endurances of annealed GeSAs devices. The lifetime of GeSAs$_{25}$ and GeSAs$_{43}$ devices is prolonged after annealing.

the pure GeS device fails, but GeSAs devices can still work normally and so do devices with different sizes, as shown in Fig. 2b (200 nm) and Fig. S9 (60 nm). In Fig. 2b, $V_h$ hardly changes, but $V_{th}$ of annealed GeSAs$_{20}$ fluctuates between 2.1 and 2.9 V, while the function of As on performance uniformity is still effective after annealing, and the $V_{th}$ fluctuating value of GeSAs$_{43}$ is only 0.24 V (1.73 V–1.97 V). Moreover, the DC test circuit is consistent with that before annealing, and $V_{fire}$ of annealed GeSAs devices slightly increase to 4.3 V–4.4 V. After the FF process as described by dashed lines, $V_{th}$ of GeSAs$_{20}$ lies in the range of 3.2 V–3.8 V (solid lines), while a sudden transition of the current occurs in GeSAs$_{25}$ devices at 2.7 V–3.3 V. Furthermore, $V_{th}$ decreases to 2.2 V–2.6 V with 43 at.% As incorporation. $V_{th}$ decreases with increasing As content, as shown in Fig. S10. At the same time, the changing trend of $I_{off}$ in GeSAs devices is also similar to that before annealing, i.e., $I_{off}$ goes up with the As concentration from 20 at.% to 43 at.%, as illustrated in Fig. 2c. $I_{off}$ of GeSAs$_{20}$ is still the smallest of the three components, reaching 3 nA, then rises up to 38 nA when the content of As is 25 at.%. and continues to 58 nA in GeSAs$_{43}$ devices, basically in line with the statistical law of $I_{off}$ and $I_{on}$ of different devices with As content shown in Fig. S10. As displayed in the figure, the average $I_{on}$ drops from 0.71 mA to 0.7 mA and further to 0.41 mA with 43 at.% As incorporation.

Furthermore, we compare the performance variations of GeSAs devices with different thicknesses of the functional layer before and after annealing, as shown in Fig. 2e (taking GeSAs$_{43}$ devices as an example). The performance-changing tendencies of all GeSAs devices with various thicknesses are shown in detail in Figs. S11–S14. In Fig. 2e, it is obvious that $V_h$ is thickness-independent[17], yet $V_{fire}$ and $V_{th}$ seem to increase nonlinearly as the thickness doubles, no matter whether the devices are annealed or not. Although $I_{on}$ increases with thickness, $I_{off}$ decreases owing to the smaller conductance caused by increasing thickness for every As content, resulting in a larger selectivity ($I_{on}/I_{off}$). A large selectivity value of >10$^5$ can be achieved in 20 nm-thick GeSAs$_{43}$ devices. However, the value of $I_{on}$ after annealing is generally lower than that before annealing, which is probably due to oxidation of the top TiN electrode. After annealing at 450 °C, GeSAs$_{20}$ devices operate normally for each pulse after 10$^7$ cycles (Fig. S15 and Fig. 2f). In the case of GeSAs$_{25}$ devices, repeated operations can reach 10$^9$ cycles (Fig. S15 and Fig. 2f), whereas the GeSAs$_{43}$ cell can be switched on and off by each pulse for a remarkable 9 × 10$^9$ cycles (Fig. S16 and Fig. 2f). These results prove that As incorporation effectively prolongs the device lifetime of the OTS selectors. Compared with reported OTSs, annealed GeSAs devices present a better overall performance, as shown in Table 1.

As we can see from the table, AsTeGeSiN device still operates normally after annealing at 500 °C for 15 minutes, which is the highest heat-treatment temperature[30]. However, $I_{off}$ of the annealed device is only 0.2 μA, which leads to a rather low storage density[30,31]. Similarly, the $I_{off}$ of the GeSe-based OTS material is only 1 μA and its on/off ratio is 10$^3$, which is the lowest among these materials[32]. Compared to

GeSe, the leakage current of TeAsGeSiSe is 5 nA, but $J_{on}$ drops to 0.44 MA/cm$^2$, that is insufficient to drive the PCM[33]. Similar to TeAsGeSiSe, the on-state current density of annealed Ge-Se-Sb-N device is only 0.2 MA/cm$^2$, although its $I_{off}$ is as low as 0.1 nA and the selective ratio is as high as 10$^6$ [34]. CTe combines a $J_{on}$ of 11 MA/cm$^2$ and a nA-scale $I_{off}$, while the device endurance decays from 10$^8$ to 10$^6$ cycles[31]. In fact, 3D PCM requires comprehensive performance of OTS materials, so we visualize the performance of these materials from five perspectives: thermal stability, endurance, $J_{on}$, $I_{off}$, and selectivity as shown in Fig. 3. Evidently, $I_{off}$ of annealed NGeCTe, GeSAs$_{25}$, and GeSAs$_{43}$ devices are relatively low, and exhibit high $J_{on}$ without sacrificing the device endurance. However, GeSAs$_{25}$ and GeSAs$_{43}$ devices deliver larger $J_{on}$ and lower $I_{off}$ with relatively high lifetime, revealing higher competitive than NGeCTe[35]. However, a higher As content will lead to a decrease in $V_{th}$, which almost overlaps with $V_h$ and squeezes the read margin.

Besides, the experimental results demonstrate that moderate As incorporation could significantly reduce the leakage current and suppress the $V_{th}$ drift, and, most importantly, it strongly enhances the thermal stability of OTS materials, improving the switching repeatability and prolonging the device lifetime, therefore enabling a processing-line-compatible OTS selector with superior properties for 3D memory applications. Based on the above results, GeSAs$_{25}$ is the optimal component.

## Discussion

Yet, what are the hidden mechanisms for these performance enhancements upon As incorporation? In order to reveal the physics of OTS behavior and the important role played by As, we performed ab initio molecular-dynamics (AIMD) simulations based on density-functional theory (DFT). Models of amorphous GeS (a-GeS) and GeSAs (a-GeSAs), as presented in Fig. 4a, were generated by using a melt-quench-relaxation method. We first analyzed the number of valence electrons and the charge transfer between different elements in those amorphous GeSAs systems using the Bader Charge code[36–38], as shown in Fig. 4b. The average charge transfer for these elements was found to be −0.72 (Ge) and +0.72 (S) in a-GeS, and −0.66 (Ge), +0.71 (S) and −0.09 (As) in a-GeSAs$_{20}$. These values do not change a lot in a-GeSAs$_{25}$ (Ge: −0.63, S: +0.70, As: −0.10) and in a-GeSAs$_{43}$ (Ge: −0.53, S: +0.65, As: −0.07) compared with those for a-GeSAs$_{20}$. Interestingly, the electron transfer of S atoms in a-GeS shows a bimodal distribution, e.g., S atoms receive electrons with the numbers of either -0.4 or -0.8. These two models correspond to two major configurations of chemical environment, e.g., homopolar bonds and heteropolar bonds, as shown in Fig. S17. More interestingly, As atoms appear to be almost electroneutral, showing both positive and negative small values of charge transfer, which is because As can bond with all Ge/As/S atoms, as listed in Fig. S17. This can be rationalized by the fact that As belongs to group VA in the periodic table, possessing five outer valence electrons, and lies exactly between groups IVA (where Ge resides) with four valence electrons and VIA (where the chalcogens S/Se/Te reside) with six

**Table 1 | Summary of ovonic threshold-switching device performances after annealing using different materials**

| Material | Feature Size (nm) | Thickness (nm) | Selectivity | $J_{on}$ (MA/cm$^2$) | $I_{off}$ (A) | $V_{th}$ (V) | $V_h$ (V) | Speed (ns) | Endurance | Thermal stability |
|---|---|---|---|---|---|---|---|---|---|---|
| GeSAs$_{20}$ | 60 | 10 | 10$^5$ | 21 | 2.3 × 10$^{-9}$ | 3.3 | ~1.9 | ~10 | 10$^7$ | 450 °C/30 min |
| GeSAs$_{25}$ | 60 | 10 | 10$^4$ | 16 | 1.2 × 10$^{-8}$ | 2.5 | ~1.5 | ~10 | 10$^9$ | 450 °C/30 min |
| GeSAs$_{43}$ | 60 | 10 | 10$^4$ | 12 | 1.5 × 10$^{-8}$ | 2 | ~1.4 | ~10 | ~10$^{10}$ | 450 °C/30 min |
| NGeCTe[35] | 32 | 15 | 10$^4$ | 12 | ~2 × 10$^{-8}$ | ~1.5 | ~1.2 | — | — | 400 °C/30 min |
| AsTeGeSiN[30] | 30 | 40 | 10$^3$ | 11 | ~ 2 × 10$^{-7}$ | ~1.5 | — | — | — | 500 °C/15 min |
| GeSe[32] | 50 | 5 | 10$^3$ | — | 10$^{-6}$ | ~1.4 | ~0.5 | 2 | — | 350 °C/4 min |
| TeAsGeSiSe[33] | 350 | 20 | 10$^4$ | 0.44 | 5 × 10$^{-9}$ | 3 | — | — | — | 350 °C/30 min |
| GeSeSbN[34] | 350 | — | 10$^6$ | 0.2 | 10$^{-10}$ | 2.5 | ~1.3 | — | 10$^8$ | 400 °C/30 min |
| CTe[31] | 30 | ~10 | 10$^5$ | 11 | 5 × 10$^{-9}$ | ~0.6 | ~0.3 | <10 | 10$^6$ | 450 °C/30 min |

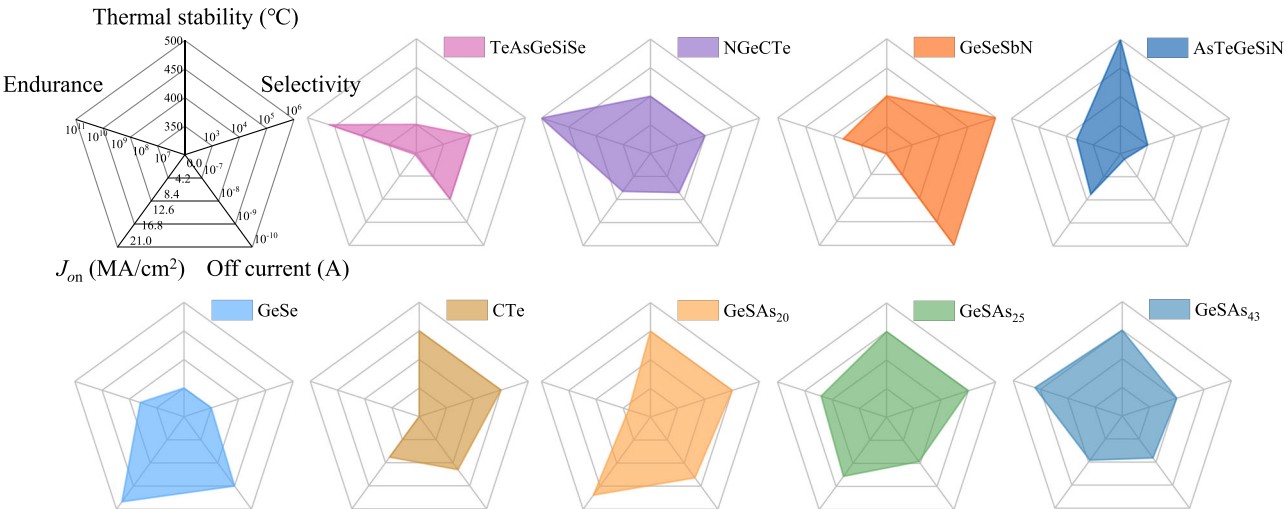

**Fig. 3 | Radar charts of different materials.** The larger the area of the figure, the closer it is to the regular hexagon, indicating that the properties of the material are superior all around. Endurances of annealed NGeCTe, AsTeGeSiN, GeSe, TeAsGeSiSe devices, and the $J_{on}$ of GeSe OTS are evaluated based on their as-deposited performance.

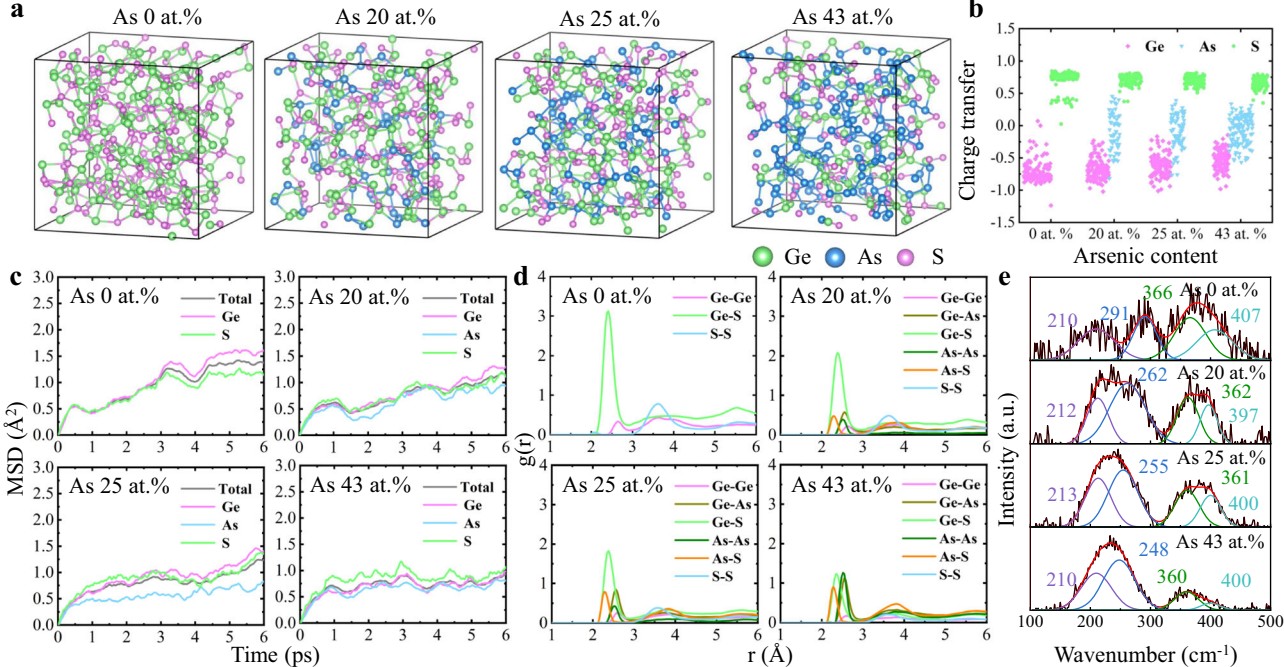

**Fig. 4 | Structural models, electronic charge transfer, atomic-mobility properties, and Raman-scattering results for amorphous GeS, GeSAs$_{20}$, GeSAs$_{25}$, and GeSAs$_{43}$. a** Snapshots of the model atomic structures after full relaxation. **b** The calculated charge transfer for the three elements, in which the ±signs represent gain/loss of electrons, respectively. **c** Mean-square displacement (MSD) of atoms in these models at 600 K for 6 ps. **d** Partial PDFs, $g(r)$, simulated at 300 K. **e** Experimental Raman spectra with Gaussian fitting of peaks.

valence electrons. This is quite different from other dopants in a-GeSe such as C[35,39], Si[40], N[39,41], B[42], I[10] et al., acting as either cations or anions, which may be responsible for serious side-effects, like high $I_{off}$ (C doping)[39], higher $V_{fire}$, $V_{th}$(N doping)[41] or poor thermal stability (B doping)[42]. The electrically neutral nature of As makes Ge-S/Se/Te chalcogenides tolerate higher concentrations of As without destroying the system's electroneutrality. From this perspective, phosphorus (P) in group VA may also serve as a promising doping candidate for enhancing the overall performance of Ge-S/Se/Te OTSs.

We evaluated the extent of atomic migration by calculating the mean-square displacement (MSD) for the models to identify the effect of As incorporation on the kinetic properties, which are usually linked with the stability of glass. The calculated MSD at 600 K in Fig. 4c shows that the total atomic movements in a-GeSAs$_{20}$, a-GeSAs$_{25}$, and a-GeSAs$_{43}$ are about 50% slower than in a-GeS, and the As atoms are the slowest in all a-GeSAs models. These results indicate that As could slow down the total atomic mobility due to the presence of relatively stronger As-S bonds (as confirmed by ICOHP, a bond-strength indicator shown in Fig. S18), namely, the incorporation of As hinders the atomic migration. It is noteworthy that the MSD is positively correlated with the diffusion coefficient ($D$) of amorphous solids or glasses. Smaller values of MSD correspond to smaller values of $D$ and larger activation energies ($E_a$) since $D \sim \exp(-E_a/k_BT)$[43], thereby accounting for the >100 °C increase in the crystallization temperature after As

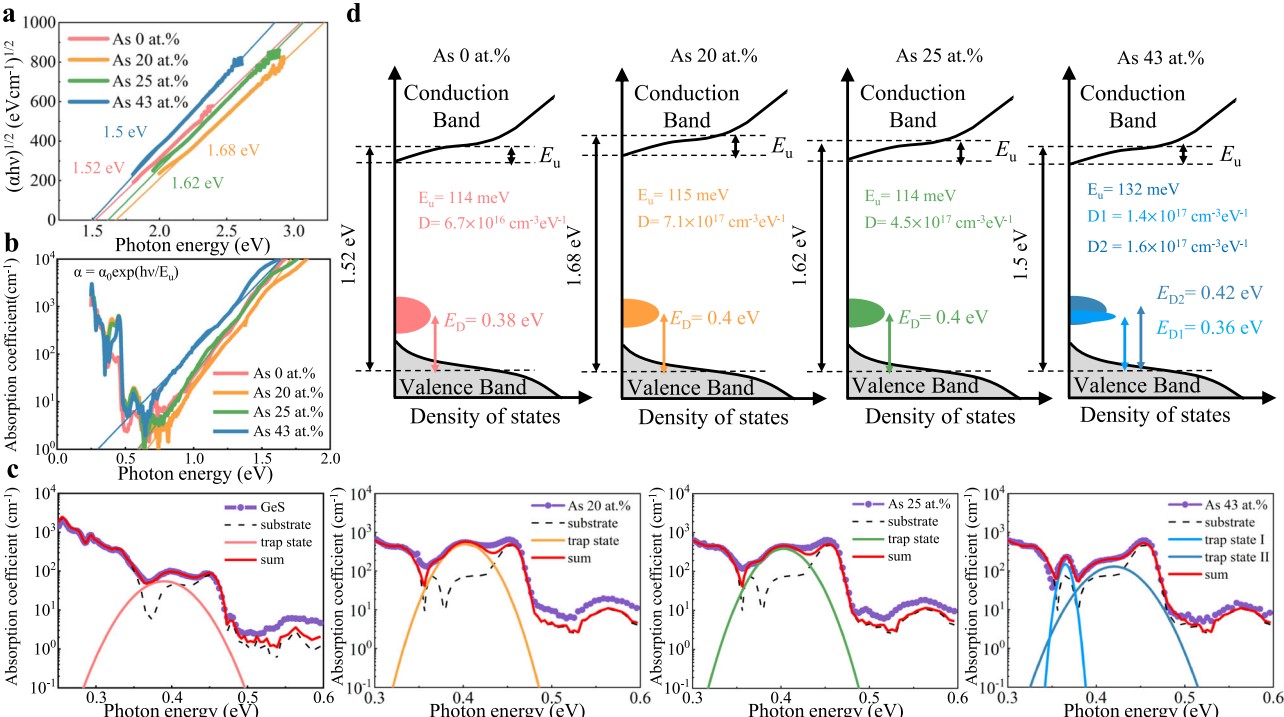

**Fig. 5 | Trap states and bandgaps of amorphous GeSAs films. a** Tauc plots of the optical-absorption coefficient, $(\alpha h\nu)^{1/2}$ versus $h\nu$, provide estimates for the bandgap. **b** Semi-logarithmic plots of $\alpha$ versus $h\nu$ characterize the Urbach tail, where $\alpha_0$ is independent of either thermal or structural disorder, while $E_u$ is the Urbach-edge parameter. **c** In-gap trap states detected by PDS spectra; an extra one appears for 43 at.% As. **d** Experimentally determined energy-band diagrams of amorphous GeSAs.

incorporation, as illustrated in Fig. 2a and Fig. S7. Together, the high crystallization temperature (>500 °C) and the low atomic mobility in amorphous GeSAs make it possible to achieve ~$10^{10}$ cycling endurance, as shown in Fig. 2f.

Normally, due to the stronger electronegativity of sulfur atoms, it is more likely to form Ge-S heteropolar bonds than Ge-Ge homopolar bonds, which is demonstrated by different intensities of peaks in the PDF located at ~2.38 and ~2.61 Å in the a-GeS models, as shown in Fig. 4d. The dominance of Ge-S bonds in a-GeS can also be confirmed by the presence of the major peaks at 210 and 407 cm$^{-1}$ in the Raman spectra (Fig. 4e), which are associated with Ge-S chains[17]. The vibrational modes of Ge-Ge and Ge-S bonds in the molecule S$_3$Ge-GeS$_3$ are located at 291 and 366 cm$^{-1}$, respectively[17]. Nonetheless, owing to being surrounded by large numbers of Ge atoms (41.8% of the next-nearest neighbors, Fig. S19), Ge atoms could migrate and form Ge-Ge filaments[44,45] triggered by high electric fields in the FF process[17,46,47], leading to the delocalization of conduction state. Such paths dramatically increase the conductivity, resulting in a sharp increase in $I_{off}$ by more than 100 times in the following switching operations (Figs. 1b and 2c). Yet, these Ge-Ge paths would be slowly dissolved due to their instability without external electric field after removal of the voltage[17,21], $V_{th}$ thereby spontaneously increases[48] causing the so-called $V_{th}$ drift, observed in Fig. 1e.

After the incorporation of As, As-S, As-As, and Ge-As bonds emerge, and peaks in the PDF can be identified with such atom-atom correlations, located at 2.28, 2.51, and 2.55 Å, respectively (Fig. 4d)[49,50]. At the same time, the amplitudes of the Ge-Ge and Ge-S peaks in the PDF decrease as well, indicating that As atoms could bond with all Ge/S/As atoms. Similar conclusions can be drawn from experimental Raman results (Fig. 4e): As-S vibrations in As$_4$S$_4$[51,52] and interactions between AsS$_3$ pyramids[52] appear at 362 and 397 cm$^{-1}$, while peaks at 212 and 262 cm$^{-1}$ can be attributed to GeS chains[17] and ethane-like S$_3$Ge-GeS$_3$ units[53], when there are still a great deal of Ge-S bonds at the point of As 20 at.%. Similarly, with a further increase of As to 25 at.%, Raman

peaks at 361 and 400 cm$^{-1}$ correspond to As$_4$S$_4$[51,52] and AsS$_3$[52] units, while the 213 and 255 cm$^{-1}$ Raman peaks are ascribed to Ge-S chains[17] and S$_3$Ge-GeS$_3$[54] as well, indicating that the Ge-S bonds still dominate at this time. However, the domination of Ge-S bonds is replaced in GeSAs$_{43}$ where the number of As-As bonds emerge in abundance at 248 cm$^{-1}$[55], while other peaks at 210, 360, and 400 cm$^{-1}$ stand for Ge-S chains[17], As$_4$S$_4$[51,52] and AsS$_3$[52] atomic groups. As a result, the proportion of Ge atoms that are next-nearest neighbors of other Ge atoms significantly falls to 30.7% for GeSAs$_{20}$ and further to 20.3% for GeSAs$_{43}$ (Fig. S19), thereby decreasing the possibility of generating long Ge-Ge filaments through diffusion. These results together with the slow atomic migration inhibited by As (Fig. 4c) account for the slower $V_{th}$ drift, as observed in Fig. 1e.

The electrical conduction in the sub-threshold region of OTSs is believed to be controlled by the Poole-Frenkel mechanism, that is, with charge carriers hopping from one trap to the conduction band and then captured by another trap[25,26]. Thereby, the decrease at first and then an increase of $I_{off}$ with increasing As incorporation can be explained in terms of the width of the bandgap and the density of trap states of amorphous GeSAs films, which can be characterized by photothermal deflection spectroscopy (PDS) experiments. As obtained from Tauc plots of such data (Fig. 5a), the bandgap increases from 1.52 eV to 1.67 eV with an increase of the As content from 0 at.% to 20 at.%, and then a decrease to 1.62 eV, ending up at 1.5 eV for GeSAs$_{43}$. This shows the same trend as the variation of $I_{off}$ with different compositions (Fig. 1b, c), and it is also consistent with the compositional tendency of the conductivity activation energy (Fig. S20). The downward trend probably results from the Mott delocalization caused by an increase in the concentration of As-As bonds[56,57], which have been evidenced in the theoretical PDFs from modeling studies (Fig. 4d) and further confirmed by experimental Raman results (Fig. 4e).

Since the sample absorbs the excitation light in the PDS measurement, not only does it generate heat, but it also causes the carriers to transition between valence band and defect state, defect state and

conduction band, and even different defect states, resulting in additional photo absorption. Therefore, the trap-state positions shown in Fig. 5c can be obtained by Gaussian fitting of the absorption curves, from which we observed that trap states in a-GeS are located at 0.38 eV above the valence-band maximum. In samples with 20 and 25 at.% As contents, the trap states are located at 0.4 eV. However, two defect states appear in the GeSAs₄₃ sample; they occur at 0.36 and 0.42 eV respectively, presumably due to the formation of As clusters, as shown in Fig. 4d, e. It should be noted that more trap states in a-GeS were detected in a previous work[17], mainly due to the overlap of the absorption associated with the traps and the deflection medium, as detailed in Fig. S21. The Urbach tail energy ($E_u$) in Fig. 5b is obtained by a linear fitting of the absorption-coefficient data, $\alpha$, with photon energy, $h\nu$. Summing up all the information outlined above yields experimentally determined energy-band diagrams, as displayed in Fig. 5d. The trap densities can be calculated from the intensities of the absorption peaks, which are estimated to be $6.7 \times 10^{16}$, $7.1 \times 10^{17}$, and $4.5 \times 10^{17}\,\mathrm{cm^{-3}eV^{-1}}$ for trap D in pure GeS, GeSAs₂₀, and GeSAs₂₅, respectively. The trap densities of D1 and D2 in GeSAs₄₃ are $1.4 \times 10^{17}$ and $1.6 \times 10^{17}\,\mathrm{cm^{-3}eV^{-1}}$. The trap densities show a trend of firstly increasing and then decreasing with the increasing As concentration, while the turning point is situated at 20 at.% As, the same trend as observed for the bandgap (Fig. 5b). According to the Poole-Frenkel mechanism[25,26], more traps imply that more carriers generated by an excitation signal would be captured. Moreover, a larger bandgap also leads to a larger energy barrier ($E_C - E_D$) between the trap state and the conduction band[25,26]. Both factors result in the increasing and decreasing trend for $I_{off}$, as shown in Fig. 1b, e.

The nature of these trap states can be identified from further analysis of the DFT models, as presented in Fig. 6. The calculated electronic density of states (DOS) and corresponding normalized inverse participation ratio (IPR) of a-GeS, a-GeSAs₂₀, a-GeSAs₂₅, and a-GeSAs₄₃ are shown in Fig. 6a. In general, larger IPR values indicate more strongly localized electron states. We determine the mobility gap ($E_g$) of the amorphous models by calculating the energy separation between the mobility edges, defined by relatively lower IPR values of valence- and conduction-band states compared with the trap states in the bandgap[58]. The values of $E_g$ for a-GeS, a-GeSAs₂₀, a-GeSAs₂₅, and a-GeSAs₄₃ calculated by hybrid potential functionals are 1.55, 1.70, 1.59, and 1.48 eV, respectively in line with the experimental results shown in Fig. 5a. All the DOSs of the models exhibit evident trap states marked as A, B, C, D, and E in the mobility gaps, which are located at 0.38, 0.42, 0.41, 0.33, and 0.45 eV above the valence-band mobility edges, respectively, which is consistent with the experimental results shown in Fig. 5c. All the trap states show large IPR values, indicating that the carriers trapped at these localized states will contribute little to the electrical conduction at room temperature because of their low mobility. However, the energy profile associated with the mobility gap could be tilted when a voltage bias is applied, leading to the tunneling of carriers from trap states to the valence bands. Besides in-gap states, the IPR values are usually large near the tail of the conduction band where the electronic states are strongly localized too (Anderson localization), as shown in Fig. 6a and Fig. S22[59,60]. In order to find the origin of the trap states, we projected them onto real space by using the analytical tools for electron wave functions in the VASPKIT code[61], as shown in Fig. 6b, c. The trap states A, B, C, and D in the a-GeSAs models are mainly found to be associated with structural motifs consisting of Ge-Ge bonds/chains[44,45,59,62], while the trap state E of a-GeSAs₄₃ is different from the others, and it is dominantly associated with As-As bonds/chains. As the major sources of these traps, Ge-Ge bonds/chains play a crucial role in OTS behavior, and As/S atoms also contribute to these in-gap states from the PDOS. Compared with trap A

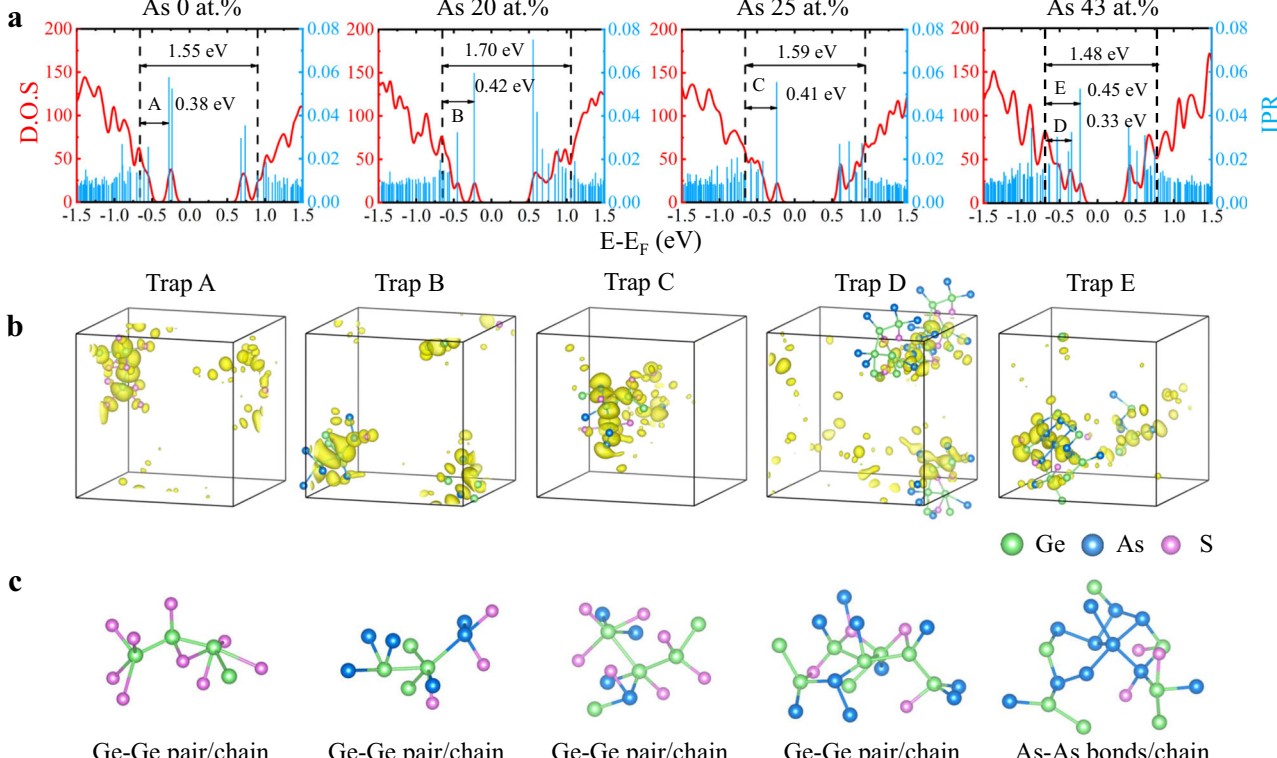

**Fig. 6 | First-principles simulations of the trap states for a-GeSAs systems.**
**a** Density of states (DOS) and normalized inverse participation ratio (IPR) of a-GeS, a-GeSAs₂₀, a-GeSAs₂₅, and a-GeSAs₄₃, which show the mobility gap and in-gap states. **b** The trap states projected onto real space, in which the yellow areas, plotted with isovalues of $1.3 \times 10^{-10}\,\mathrm{e \times bohr^{-3}}$ encapsulate all the in-gap states and depict the molecular-orbital-charge density of the respective trap state. **c** The atomic clusters associated with the trap states.

in a-GeS, As atoms participate in the contribution of trap states in the range of low As-doping contents, leading to an increasing density of in-gap states from $6.7 \times 10^{16}$ $cm^{-3}eV^{-1}$ to $7.1 \times 10^{17}$ $cm^{-3}eV^{-1}$, as shown in Fig. 5d. Interestingly, the contribution of Ge-Ge chains to the trap states appears to become saturated when the As content reaches 20 at.%, thereafter the total density of in-gap states starts to decrease, even though the As-As bonds provide extra traps (Fig. 5) which originate from As separation. This exactly explains the non-monotonic effect of As incorporation on $I_{off}$ as the As content increases.

In conclusion, we have investigated OTS devices made of GeS, $GeSAs_{20}$, $GeSAs_{25}$, $GeSAs_{43}$ materials and studied the influence of As content on the key OTS performance parameters through electrical measurements. Annealing tests at 450 °C for 30 minutes, that mimic the industrial BEOL processing, confirm that materials containing As show higher crystallization temperatures than pure GeS, and the devices can be still operated normally after such harsh heat treatment, indicating that the incorporation of As could remarkably enhance the thermal stability. A significantly slower drift of $V_{th}$ and a better device lifetime are also observed. This is because As atoms form strong bonds with both Ge and S, which also slows down the atomic migration, as confirmed by DFT-simulation calculations and experimental Raman spectra. In addition, the incorporation of As improves the OTS performance by modifying the bandgap and trap states. The trap states in the energy-bandgap, which are the key feature that leads to the OTS behavior, are enhanced due to the presence of Ge-As and As-S bonds. In particular, new trap states are found in $GeSAs_{43}$, mainly because the excess As atoms can induce As segregation, in which homopolar As-As bonds generate extra free electrons. Interestingly, As atoms appear to be almost electroneutral, which is likely to be the reason that the serious side-effects induced by other dopants can be avoided and this enables an excellent overall selector performance for practical switching in memory-array applications. Our work aims to understand the mechanism of As doping in the newly developed GeS selector, thereby paving the way for the optimization of new 3D PCM products.

## Methods

### Device preparation and measurement
The 10 nm-GeSAs layers of all components were RF-sputtered, utilizing GeS, $(GeS)_{80}As_{20}$, $(GeS)_{75}As_{25}$, and $(GeS)_{57}As_{43}$ alloy targets using a power of 25 W. The 5 nm C layers and top TiN electrodes were deposited by DC-sputtering using powers of 40 and 75 W, respectively. The device performance was characterized by a Keithley 4200A-SCS instrument. The device performance was characterized by applying pulses through a Keithley 4200A-SCS instrument rather than DC test because of the significant damage to the device caused by DC test. Figure 1c was obtained by applying 100 consecutive 3, 4, 3.5, and 3.5 V triangular pulses to GeS, $GeSAs_{20}$, $GeSAs_{25}$, and $GeSAs_{43}$ as-deposited devices initialized by a 6.5 V triangular pulse for $GeSAs_{20}$ and a 6 V one for other three. Similarly, the pulses used to fire the annealed devices in Fig. 2b are 8.7, 7, and 7 V for $GeSAs_{20}$, $GeSAs_{25}$, and $GeSAs_{43}$. The 100 pulses for operation are 5, 4, and 4 V, respectively. As for $V_{th}$ drift, a combination of a high and a low triangular pulse with 1 μs interval was applied firstly to the device. The higher one is used to fire the device, while the lower one is used to measure the $V_{th}$ and we took the moment that the lower pulse was input as zero point. The amplitudes of pulses for firing are 6, 6.5, 6, and 6 V with the increasing As concentration. And the testing pulses are 3, 4, 4, and 4 V. In Fig. 2d, as-deposited $GeSAs_{43}$ devices with the OTS layer thicknesses ranging from 5 nm to 20 nm are fired by a 5, 6, and 9 V triangular pulse and operated by a 3, 4, and 5.5 V one, respectively. A 5, 7, and 9 V triangular pulse is used to fire the annealed devices from 5 nm–20 nm thickness. $V_{th}$ was obtained by a 3, 4, and 5 V pulse for each. All the rising and falling edge periods of the triangular pulse are 1 μs. Protocols of detailed measurements in supplement material are mentioned in their captions.

### Structure and band-structure characterization
All of the samples used for Raman scattering were 100 nm thick. A Renishaw inVia Qontor Raman microscope with a laser-excitation wavelength of 532 nm was utilized to obtain the Raman spectra, and the samples for PDS measurements were about 400 nm thick and deposited on fused quartz substrates. The Raman peaks may have some deviation from the literature results due to the different chemical environment, but the deviation is less than 3 $cm^{-1}$. Excitation light for the PDS measurements came from a 100 W tungsten halogen lamp with a monochromator (CM110). The deflection signal was detected by a position-sensitive sensor (S3979, Hamamatsu Photonics). The samples were immersed in a liquid for the sake of signal enhancement. Tauc plots and the photothermal deflection spectra were obtained extending to a wavelength of 5000 nm. The Urbach tail energy ($E_u$) in Fig. 4b was obtained by linear fitting of the absorption coefficient, $\alpha$, to the photon energy, $h\nu$, using the equation $\alpha = \alpha_0 exp(h\nu/E_u)$, where $E_0$ refers to the optical gap, and the resulting widths of the localized band tails were found to be 114, 115, 114, and 132 meV below the conduction-band minimum for As contents of 0, 20, 25, and 43 at.%, respectively.

### Cs-corrected TEM characterization
TEM samples were fabricated by focused ion beam (FIB) milling. The images for TEM and EDS analysis were taken using a JEOL JEM-ARM300F microscopy.

### Atomic-model simulations
The Vienna Ab initio Simulation Package (VASP) code was adopted to perform first-principles calculations[63,64]. The projected augmented-wave (PAW) method was used and the Perdew-Burke-Ernzerhof generalized-gradient approximation (GGA-PBE)[65,66] or a hybrid (HSE06) functional[67–69] were employed to describe the exchange and correlation of the electrons. AIMD simulations based on DFT were performed to generate amorphous models by using a melt-quench-relaxation method. Each supercell contained 300 atoms, and the time step used was 3 fs and the cutoff energy of the plane-wave basis was set to 300 eV in the AIMD simulations. We initially built the GeSAs supercells by randomly putting Ge, S, and As atoms into boxes with the experimentally determined density, and fully melting the cells at 3000 K for 30 ps. The liquid phase was then cooled down to 300 K at a fast rate of 30 K per picosecond, and then equilibrated at 300 K for 30 ps. These models were further relaxed at 0 K to calculate the electronic structures. All atoms were relaxed with Γ-point sampling of the Brillouin zone until the atomic forces on each atom were smaller than 0.001 eV $Å^{-1}$ and the energies were converged to $1 \times 10^{-6}$ eV. The cutoff energy was set to 500 eV and the ionic and electronic convergence precisions were $10^{-6}$ and $10^{-7}$ eV, respectively. The cubic-box sizes of the relaxed amorphous models were 19.75, 19.97, 20.10, and 20.31 Å for a-GeS, a-$GeSAs_{20}$, a-$GeSAs_{25}$, and a-$GeSAs_{43}$, respectively.

## Data availability
The data that support the findings of this study are available from the corresponding author upon reasonable request.

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

## Acknowledgements

Financial support was provided by the Strategic Priority Research Program of the Chinese Academy of Sciences (XDB44010200). M. Zhu acknowledges support from the National Outstanding Youth Program (62322411), the Hundred Talents Program (Chinese Academy of Sciences), and the Shanghai Rising-Star Program (21QA1410800). M. Xu acknowledges the National Key R&D Plan of China (2022ZD0117600) and the National Natural Science Foundation of China (62174060). T. Gotoh acknowledges the JSPS KAKENHI (21K04861).

## Author contributions

R.W. and S.J. deposited the film, prepared the devices, and measured device performances. T.G. performed the PDS experiments and analyzed the results. R.G. and M.X. performed the DFT calculations. M.Z., R.W., R.G., M.X., and S.R.E. wrote the paper. Z.Z., Y.S., X.M., and Z.S. discussed the results and commented on the manuscript. The project was initiated and conceptualized by M.Z.

## Competing interests

The authors declare no competing interests.
