## [Peer Review File · Nature Communications]

REVIEWER COMMENTS

Reviewer #1 (Remarks to the Author):

The authors provide an interesting analysis of GeAsS alloys integrated in analytical OTS devices for preliminary electrical evaluation, supported by material analyses and by AIMD simulations. Please find in the following some comments/remarks to be addressed to improve the paper quality. As a general comment, what is missing at the end is a clear guideline to optimize the As content in GeAsS system. Is coming from simulations? Or from devices results? There are some trends, but not clearly correlated one to the other. We can conclude that As is responsible for a "reduced mobility" due to Ge-As and As-S formation, however it could form As clusters: it could be detrimental? Moreover, the paper title should be revised and focused on the real objective of the work: As content effects in GeS_xAs_{1-x} system.

- 1) Line 30: "... incorporation of 20 at. % As brings a more than 100 °C increase in crystallization temperature...". This could be not valid for all the alloys. It would be better to specify since the abstract the alloy investigated.
- 2) Line 42: "These findings allow the use of precise control of the As concentration in OTS materials for improved-performance high-density 3D PCM applications". Again, it would be better to restrict to the alloy investigated: GeS+As.
- 3) I think the title is too much generic, while the authors refer to a specific alloy investigation. As a reader, I would prefer to find better specified in the title the addressed system of the work, since the As effects strongly depends on its arrangement in the system and its bonds with the other elements.
- 4) The quality of the picture is not suitable for publication. Please, improve the pictures quality.
- 5) In Fig. S7 the authors highlight a peak for GeS at 30°. Such peak is not indexed in standard databases for GeS. Which is the reference used? It is quite difficult to support by XRD results the improved stability by As introduction. Are the y-axis using the same scale?
- 6) In Fig. S8 the ON current is defined as the current induced in the device at the threshold event. Such parameter is not of real interest since it depends on the threshold voltage and on the ON resistance of the device. The ON resistance is related to the series external resistance only? If yes it should be specified that the graphs report the IV characteristics of the device in series with some external resistance.
- 7) In Fig. S9 in the graph the fail for GeS is reported to be at 400°C, while in the caption the authors specify an annealing at 450°C 30min. Which is the real thermal budget applied?
- 8) Line 140-150: the ON current calculation seems more dependent on the external resistance used than from the material local parameters (i.e. On current equal to V_{th} over ON resistance). It would be better to clarify if the author is referring more to the V_{th} of the material than to the ON current.

9) Line 145: since V_{th} could vary dependently on ramp rate used, a variation between AC and DC protocols is expected. Data in S2 etc. are obtained from DC or AC protocol? It would be better to specify it each time to avoid misunderstanding, and explain why this choice in the experimental description.

10) Line 156: how the authors take into account the RC times? The ON resistance of the device is not calculated. How much does it affects the switching on/off speed? Since no trend is observed in S4d it would be better either to remove this calculation or to better support the result with the parasitic analysis of the used equipment.

11) Data from Fig.1e cannot be read. Moreover, there is no statistics. Is there a real trend increasing As content? Is this trend confirmed in cycled (and after how many cycles?) devices?

12) The reported endurance tests are interesting. However, what it is missing is a statistics to validate the observed trend.

13) In the endurance plots, we can observe a clear evolution of the ON resistance of the device along cycling, and even if bottom right IV plots of S14 and S15 are not perfectly clear, they seem to reveal an evolution of the material along cycling. How the atomic migration/composition evolution in the formed filament/channel can be excluded (line 275)? A TEM image after the endurance test would have been interesting to validate such hypothesis, since in literature several proofs of phase separation in such system have been previously reported.

14) Line 281: looking at the current densities implied in the switching mechanism it is reasonable to expect temperatures in the ON region of the device well higher than 500°C. What could be interesting is to validate a low crystallization speed thanks to "As" introduction, since the high crystallization temperature could hinder crystallization phenomena along the BEOL thermal budget of the fabrication, but its correlation with endurance is not direct. Maybe the authors could better explain such link.

15) Fig.3e should report either in the caption, either in the graph the indexation of the deconvoluted peaks.

16) Line 336: the Raman analysis evidencing As-As vibration seems not reported in the description.

17) Line 346: As-As clusters formation in Fig.3d and 3e is not clear/evidenced. Clusters formation can also be interpreted as a segregation phenomenon. How such clusters should evolve during the device operations?

18) Raman spectra were performed on as-dep layers: it would have been interesting to perform same analyses on annealed samples, to observe the structural evolution at high temperature (not visible in XRD experiments).

Reviewer #2 (Remarks to the Author):

The subject of the present manuscript is a detailed study the Ovonic Threshold Switching (OTS,) of the GeSAs chalcogenide glass. The OTS behaviour corresponds to a volatile electronic commutation between highly and low resistive state when a voltage higher than the threshold voltage is applied. OTS materials are studied to replace silicon-based traditional selector technologies to enable the stacking of the memory with the selector with equivalent feature sizes, increasing indeed the density of the memory. The main challenge concerning OTS development is to find optimal compositions that allows the material to stay amorphous, when operating, but also when the thermal budget reaches above 400 °C during 3 hours.

The manuscript is well-written. The results are convincing, but for the exceptions noted below, and the insight provided will certainly prove to be of great interest well beyond the phase change material community.

OTS materials by their engineering are hard to crystallize, however conductive and stables crystallites can gradually form within the layer and lead to device failure in endurance. This is usually observed as a progressive decreasing of the OFF resistance. The authors presented in Fig.2 some nice TEM images from which resolution it is not possible to exclude the presence of nano crystals after annealing. Are the authors able to provide higher resolution measurements?

The drift phenomenon (aging) is linked to the amorphous relaxation of the material and it is often attributed, both in PCM and OTS, to the Ge-Ge bonds presence (to the fraction of tetrahedrally coordinated Ge). For the present GeSAs alloys the authors point at the formation of Ge-Ge filaments. I would be interested to know the number of atoms corresponding to an average sized filament. Would it be possible to directly observe it by TEM measurements?

The global composition of OTS is often complex in order to fulfil the specifications: high thermal stability, low leakage current, high endurance and tunable threshold voltage. To further help the reader to judge the novelty of the manuscript the performance comparison that is presented in FigS16 should not only mentioned in the main text, but better and more extensively summarized.

Although measurement protocol pulse (applied and measured pulses) can be extrapolated from Fig1b and are explained in the caption of FigS4, it would be useful if the authors could write in Methods to look for protocols in supplementary material.

Reviewer #3 (Remarks to the Author):

The authors present an interesting and detailed investigation (first-principles simulations, electrical and physical characterisation) of As-containing ovonic threshold switching materials that are currently required for selector application. They show that with increasing As content in GeAsS, the thermal stability of the material increases and leads to improved selector properties: longer endurance, better selectivity and lower threshold voltage drift. By using first-principles simulations, coupled with Raman

spectroscopy, they show that the improvements are due to As dual electron donor/acceptor property and high As-S bond strength, which makes the atomic diffusivity slower and the probability to form conductive Ge-Ge chains lowers with increasing As content. Electrical conductivity is motivated with Poole-Frenkel model and electronic properties (the mobility gap and the trap levels), that were extracted both with experimental and DFT techniques.

It is noteworthy to see how the increasing content of As atoms in the composition can explain the non-monotonic evolution of the threshold voltage, mobility gap and the related parameters. The presented understanding will enable the research community to apply the principles of electron-counting and bond-strength to continue the selector material fine-tuning and eventually come-up with an As-free OTS selector material. The conclusions are sound and supported by the theoretical and experimental findings, the theoretical methodology is state-of-the-art, which makes the work of very high quality and worth publishing. However, the novelty of the work is probably compromised, to be considered for Nature Communications, since (part of) the study was reported before (<https://doi.org/10.1016/j.scriptamat.2022.114834>).

For additional manuscript improvement suggestions, please see the commented pdf file (attached)

The Role of Arsenic in the Operation of Electrical Threshold Switches

Renjie Wu^{1,2, †}, Rongchuan Gu^{3, †}, Tamihiro Gotoh^{4, †}, Zihao Zhao^{1,2}, Yuting Sun^{1,2},
Shujing Jia⁵, Xiangshui Miao³, Min Zhu^{1, *}, Ming Xu^{3, *}, Stephen R. Elliott^{6, 7},
Zhitang Song^{1, *},

¹State Key Laboratory of Functional Materials for Informatics, Shanghai Institute of
Micro-System and Information Technology, Chinese Academy of Sciences; 200050
Shanghai, China

²University of Chinese Academy of Sciences; Beijing 100029, China

[revised manuscript text omitted]

compositions lie between 7 ns and 12 ns, and off-speeds are between 5 ns and 15 ns. A
composition-independent switching speed is mainly due to the pure electronic nature
of the OTS behavior, in which atomic migration is barely involved^{25,26}.

Other benefits of As incorporation can be found in the drift of V_{th} of devices with
time (Fig. 1e), that is, V_{th} spontaneously increases over time after the first-fire (FF)
process. The V_{th} drift could induce write/read failure in the high-density memory array,
as well as degradation of the device lifetime²⁷. For pure GeS, the V_{th} value of 1.8 V
drifts to 2.19 V within 1 hour, and then further rises to 2.3 V after 1 day. In the same
way, the V_{th} value of GeSAs₂₀ starts from 2.42 V, then increases to 2.66 V, and stops at
3 V. The rising trend of V_{th} was inhibited at 25 at. % As content, where V_{th} was around
2.5 V for all four tests, that is 2.46 V @ 0 s, 2.54 V @ 1 min, 2.5 V @ 1 hour and 2.58
V @ 1 day, i.e., only slightly increasing, while V_{th} increases from 2.28 V to 2.67 V and
further to 2.8 V @ 1 day in GeSAs₄₃. Obviously, an appropriate As content is sufficient
to restrain the V_{th} drift. For device endurance, all GeS and GeSAs devices can be
successfully turned on and off for at least 10^8 cycles, as shown in **Figs. S5-S6** and Fig. 1f.

OTS selectors must withstand a temperature of 450 °C for 30 minutes in the BEOL
process, in which the metal wire is bonded and the insulator layer is deposited^{28,29}. Since
only chalcogenide glasses exhibit threshold-switching behavior, if they transform into
a crystal under this condition, the crystallized selector then fails to exhibit any OTS
function and can no longer be recovered. We therefore studied the crystallization
temperature of Ge-S-As films utilizing X-ray diffraction (XRD) in **Fig. S7**. After
annealing at different temperatures, a crystalline peak emerges in the GeS diffractogram

after undergoing a heat treatment at 400 °C for 10 minutes, whereas three other films
containing As remained amorphous through annealing at 450 °C for 30 minutes and
even at 500 °C for 10 minutes. That is to say, incorporation of 20 at. % As brings about
a more than 100 °C increase in the crystallization temperature, directly indicating that
the incorporation of As contributes to a strong reinforcement of thermal stability. The
material morphology can be further confirmed by transmission electron microscopy
(TEM) and the corresponding fast Fourier transform images of a GeSAs₄₃ device
annealed at 450 °C for 30 minutes are shown in **Fig. 2a**. In addition, there was no
observed segregation and diffusion of elements after high-temperature treatment on the
basis of EDS mappings. We then applied 100 triangular electrical pulses to the annealed
cells, and the corresponding I - V results are shown in **Fig. 2b**. After 450 °C annealing,
the pure GeS device fails, but GeSAs devices can still work normally and so do devices
with different sizes, as shown in **Fig. 2b** (200 nm) and **Fig. S8** (60 nm). In **Fig.2b**, V_{th}
hardly changes, but V_{th} of annealed GeSAs₂₀ fluctuates between 2.1 and 2.9 V, while
the function of As on performance uniformity is still effective after annealing, and the
V_{th} fluctuation value of GeSAs₄₃ is only 0.24 V (1.73-1.97 V). Moreover, the DC test
circuit is consistent with that before annealing, and V_{fire} of annealed GeSAs devices
slightly increase to 4.3-4.4 V. After the FF process as described by dotted lines, V_{th} of
GeSAs₂₀ lies in the range of 3.2-3.8 V (dashed lines), while a sudden transition of the
current occurs in GeSAs₂₅ devices at 2.7-3.3 V. Furthermore, V_{th} decreases to 2.2-2.6 V
with 43 at.% As incorporation. V_{th} decreases with increasing As content, as shown in
**Fig. S9**. At the same time, the changing trend of I_{off} in GeSAs devices is also similar to

that before annealing, i.e., I_{off} goes up with the As concentration from 20 to 43 at.%, as
 illustrated in **Fig. 2c**. I_{off} of GeSAs₂₀ is still the smallest of the three components,
 reaching 3 nA, then rises up to 38 nA when the content of As is 25 at.% and continues
 to 58 nA in GeSAs₄₃ devices, basically in line with the statistical law of I_{off} and I_{on} of
 different devices with As content shown in **Fig. S9**. As displayed in the figure, the
 average I_{on} drops from 0.71 mA to 0.7 mA and further to 0.41 mA with 43 at.% As
 incorporation.

 **Fig.2 Microstructure and electrical performance of Ge-S-As devices with different**
 **thicknesses after 450 °C annealing. a.** TEM picture of a GeSAs₄₃ device after 450°C
 annealing. FFT images, labelled I, II and III, represent left, middle and right areas
 marked on the Ge-S-As layer, revealing an amorphous nature. EDS mappings after heat
 treatment reveal no difference with those for the as-deposited state. **b.** I - V curves of
 450 °C annealed devices subject to 100 consecutive triangular pulses. The GeS device
 fails, whereas the GeSAs ones continue to work well. **c.** DC I - V curves of 450 °C
 annealed devices. **d.** Different responses of GeSAs₄₃ devices with different thicknesses
 of the OTS layer, before and after annealing. **e.** Electrical performance for GeSAs₄₃
 devices with different thicknesses, before and after heat treatment. V_{fire} , V_{th} , V_{h} and I_{on}
 of both states increase while I_{off} monotonically decreases. **f.** Annealed devices
 demonstrate **more** superior endurance properties.

Furthermore, we compare the performance variations of Ge-S-As devices with
 different thicknesses of the functional layer, before and after annealing, as shown in Fig.

2e (taking GeSAs₄₃ devices as an example). The changing tendencies are shown in
detail in **Figs. S10-S13**. In **Fig. 2e**, the solid lines represent the deposited devices, and

[revised manuscript text omitted]

molecules⁴⁸ appear at 218 and 270 cm⁻¹, while peaks in the Raman spectra at 370 and
373 cm⁻¹ can be attributed to edge-sharing GeS₄ tetrahedra⁴⁷, when there are still a great
deal of Ge-S bonds at the point of As 20 at.%. With a further increase of As to 25 at.%,
more As-S bonds arise and contribute to the Raman peak at 218 cm⁻¹, while the 343 and
386 cm⁻¹ Raman peaks have been ascribed to vibrations of AsS₃ pyramidal units⁴⁸⁻⁵⁰.
As a result, the proportion of Ge atoms which are next-nearest neighbors of other Ge
atoms significantly falls to 30.7% for GeSAs₂₀ and further to 20.3% for GeSAs₄₃ (**Fig.**
**S19**), thereby decreasing the possibility of generating long Ge-Ge filaments through
diffusion. These results, together with the slow atomic migration inhibited by As (Fig.
3c), account for the slower V_{th} drift, as observed in **Fig. 1e**.

The electrical conduction in the sub-threshold region of OTSs is believed to be
controlled by the Poole-Frenkel mechanism, that is, with charge carriers being
thermally emitted from one trap to the conduction band and then captured by another
trap^{25,26}. Thereby, the decrease at first and then an increase of I_{off} with increasing As
incorporation can be explained in terms of the size of the band gap and the density of
trap states of amorphous Ge-S-As films, as can be characterized by photothermal
deflection spectroscopy (PDS) experiments. As obtained from Tauc plots of such data
(**Fig. 4a**), the band gap increases from 1.52 eV to 1.67 eV with an increase of the As

content from 0 to 20 at. %, and then decreases to 1.62 eV, ending up at 1.5 eV for
GeSAs₄₃. This shows the same trend as for the variation of I_{off} with different
compositions (Fig. 1b and 1c), and which is also consistent with the compositional
tendency of the conductivity activation energy (**Fig. S20**). The downward trend
probably results from Mott delocalization caused by an increase in the concentration of
As-As bonds^{51,52}, which have been evidenced in the theoretical PDFs from modelling
studies (Fig. 3d) and further confirmed by experimental Raman results (Fig. 3e).

Since the excitation light absorbed by a sample in PDS measurements not only
generates heat, but also produces a large number of electron-hole pairs, these non-
equilibrium carriers can produce additional photo-absorption centres when they are
trapped by defect states. Therefore, the trap-state positions shown in **Fig. 4c** can be
obtained by Gaussian fitting of the absorption curves, from which we observed that trap
states in a-GeS are located at 0.38 eV above the valence-band maximum. In samples
with 20 and 25 at. % As contents, the trap states are located at 0.4 eV. However, two
defect states appear in the GeSAs₄₃ sample; they occur at 0.36 eV and 0.42 eV
respectively, presumably due to the formation of As clusters, as shown in Figs. 3d and
3e. It should be noted that more trap states in a-GeS were detected in a previous work¹⁷,
mainly due to the overlap of the absorption associated with the trap positions with the
absorption of the deflection medium, as detailed in **Fig. S21**. The Urbach-tail energy
(E_u) in Fig. 4b is obtained by a linear fitting of the absorption-coefficient data, α , with
photon energy, $h\nu$. Summing up all the information outlined above yields
experimentally determined energy-band diagrams, as displayed in Fig. 4d. The trap

densities can be calculated from the intensities of the absorption peaks, which are
 estimated to be $6.7 \times 10^{16} \text{ cm}^{-3} \text{ eV}^{-1}$, $7.1 \times 10^{17} \text{ cm}^{-3} \text{ eV}^{-1}$, and $4.5 \times 10^{17} \text{ cm}^{-3} \text{ eV}^{-1}$ for trap
 D in pure GeS, GeSAs₂₀ and GeSAs₂₃, respectively. The trap densities of D1 and D2 in
 GeSAs₄₃ are $1.4 \times 10^{17} \text{ cm}^{-3} \text{ eV}^{-1}$ and $1.6 \times 10^{17} \text{ cm}^{-3} \text{ eV}^{-1}$, respectively. The trap
 densities show a trend of firstly increasing and then decreasing, with increasing As
 concentration, while the turning point is situated at 20 at. % As, the same trend as
 observed for the band gap (Fig. 4b). According to the Poole-Frenkel mechanism^{25,26},
 more traps imply that more carriers generated by an excitation signal would be captured.
 Moreover, a larger band gap also leads to a larger energy barrier ($E_C - E_D$) between the
 trap state and the conduction band^{25,26}. Both factors result in the increasing and
 decreasing trend for I_{off} , as shown in Figs. 1b and e.

 **Fig.4 Trap states and bandgaps of amorphous Ge-S-As films. a. Tauc plots of the**
 **optical-absorption coefficient, $(\alpha hv)^{1/2}$ versus hv , provide estimates for the bandgap.**
 **b. Semi-logarithmic plots of α versus hv characterize the Urbach tail, where α_0 is**
 **independent of either thermal or structural disorder, while E_u is the Urbach-edge**
 **parameter. c. In-gap trap states detected by PDS spectra; an extra one appears for 43**
 **at.% As. d. Experimentally determined energy band diagrams of amorphous Ge-S-As.**

The nature of these trap states can be identified from a further analysis of the
DFT models, as presented in **Fig. 5**. The calculated electronic density of states (DOS)
and corresponding normalized inverse participation ratio (IPR) of a-GeS, a-GeSAs₂₀,
a-GeSAs₂₅ and a-GeSAs₄₃ are shown in **Fig. 5a**. In general, larger IPR values indicate
more strongly localized electron states. We determine the mobility gap (E_g) of the
amorphous models by calculating the energy separation between the mobility edges,
defined by relatively lower IPR values of valence- and conduction-band states
compared with the trap states in the band gap⁵³. The values of E_g for a-GeS, a-GeSAs₂₀,
a-GeSAs₂₅ and a-GeSAs₄₃, calculated using hybrid potential functionals, are 1.55 eV,
1.70 eV, 1.59 eV and 1.48 eV, respectively, in line with the experimental results shown
in **Fig. 4d**. All the DOSs for the models exhibit evident trap states, marked as A, B, C,
D and E in the mobility gaps, which are located at 0.38 eV, 0.42 eV, 0.41 eV, 0.33 eV
and 0.45 eV above the valence-band mobility edges, respectively, consistent with the
experimental results shown in Fig. 4d. All the trap states show large IPR values, so that
carriers trapped at these localized states will contribute little to the electrical conduction
at room temperature because of their low mobility. However, the energy profile
associated with the mobility gap could be tilted when a voltage bias is applied, leading
to the tunneling of carriers from trap states to the valence bands. Besides in-gap states,
the IPR values are usually large near the tail of the conduction band where the electronic
states are strongly localized too (Anderson localization), as shown in **Fig. 5a** and **Fig.**
**S22)**^{54,55}. In order to find the origin of the trap states, we projected them onto real space
by using the analytical tools for electron wave-functions in the VASPKIT code⁵⁶, as

shown in **Figs. 5b-c**. The trap states A, B, C and D in the a-GeSAs models are mainly
 found to be associated with structural motifs consisting of Ge-Ge bonds/chains^{40,41,54,57},
 while the trap state E of a-GeSAs₄₃ is different from the others, and is dominantly
 associated with As-As bonds/chains. As the major sources of these traps, Ge-Ge
 bonds/chains therefore play a crucial role in OTS behavior, and As/S atoms also
 contribute to these in-gap states from the PDOS. Comparing with the trap A in a-GeS,
 As atoms participate in the contribution of trap states in the range of low As-doping
 contents, leading to an increased density of in-gap states from $6.7 \times 10^{16} \text{ cm}^{-3} \text{ eV}^{-1}$ to
 $7.1 \times 10^{17} \text{ cm}^{-3} \text{ eV}^{-1}$, as shown in **Fig. 4**.
[revised manuscript text omitted]

- 47. Mamedov, S., Georgiev, D. G., Qu, T. & Boolchand, P. Evidence for nanoscale
phase separation of stressed–rigid glasses. *J. Phys. Condens. Matter* **15**, S2397–
S2411 (2003).
- 48. Soyer-Uzun, S., Sen, S. & Aitken, B. G. Network vs Molecular Structural
Characteristics of Ge-Doped Arsenic Sulfide Glasses: A Combined Neutron/X-
ray Diffraction, Extended X-ray Absorption Fine Structure, and Raman
Spectroscopic Study. *J. Phys. Chem. C* **113**, 6231–6242 (2009).
- 49. Stronski, A. V., Vlcek, M., Tolmachovt, I. D. & Pribylova, H. Optical
characterization of As-Ge-S thin films. *J. Optoelectron. Adv. Mater.* **11**, 1581–
1585 (2009).
- 50. Yamaguchi, M., Shibata, T. & Tanaka, K. A resonance Raman scattering study
of localized states in Ge-S glasses. *J. Non. Cryst. Solids* **232–234**, 715–720
(1998).
- 51. Yuan, Z. *et al.* The enhanced performance of a Si–As–Se ovonic threshold
switching selector. *J. Mater. Chem. C* **9**, 13376–13383 (2021).
- 52. Kastner, M., Adler, D. & Fritzsche, H. Valence-Alternation Model for Localized
Gap States in Lone-Pair Semiconductors. *Phys. Rev. Lett.* **37**, 1504–1507 (1976).
- 53. Clima, S. *et al.* Ovonic Threshold Switch Chalcogenides: Connecting the First-
Principles Electronic Structure to Selector Device Parameters. *ACS Appl.*
*Electron. Mater.* **5**, 461–469 (2023).
- 54. Konstantinou, K., Mocanu, F. C., Lee, T. H. & Elliott, S. R. Revealing the
intrinsic nature of the mid-gap defects in amorphous $Ge_2Sb_2Te_5$. *Nat. Commun.*
**10**, 3065 (2019).
- 55. Slassi, A. *et al.* Device - to - Materials Pathway for Electron Traps Detection in
Amorphous GeSe - Based Selectors. *Adv. Electron. Mater.* 2201224 (2023).
- 56. Wang, V., Xu, N., Liu, J.-C., Tang, G. & Geng, W.-T. VASPKIT: A user-
friendly interface facilitating high-throughput computing and analysis using
VASP code. *Comput. Phys. Commun.* **267**, 108033 (2021).
- 57. Konstantinou, K., Mocanu, F. C., Akola, J. & Elliott, S. R. Electric-field-induced
annihilation of localized gap defect states in amorphous phase-change memory
materials. *Acta Mater.* **223**, 117465 (2022).
- 58. Kresse, G. & Hafner, J. Ab initio molecular dynamics for liquid metals. *Phys.*
*Rev. B* **47**, 558–561 (1993).
- 59. Kresse, G. & Furthmüller, J. Efficient iterative schemes for ab initio total-energy
calculations using a plane-wave basis set. *Phys. Rev. B* **54**, 11169–11186 (1996).
- 60. Blöchl, P. E. Projector augmented-wave method. *Phys. Rev. B* **50**, 17953–17979
(1994).
- 61. Perdew, J. P., Burke, K. & Ernzerhof, M. Generalized Gradient Approximation

- Made Simple. *Phys. Rev. Lett.* **77**, 3865–3868 (1996).
- 62. Chan, M. K. Y. & Ceder, G. Efficient Band Gap Prediction for Solids. *Phys. Rev.*
*Lett.* **105**, 196403 (2010).
- 63. Heyd, J., Scuseria, G. E. & Ernzerhof, M. Hybrid functionals based on a screened
Coulomb potential. *J. Chem. Phys.* **118**, 8207–8215 (2003).
- 64. Krukau, A. V., Vydrov, O. A., Izmaylov, A. F. & Scuseria, G. E. Influence of
the exchange screening parameter on the performance of screened hybrid
functionals. *J. Chem. Phys.* **125**, 224106 (2006).

REVIEWER COMMENTS

Reviewer #1 (Remarks to the Author):

The authors provide an interesting analysis of GeAsS alloys integrated in analytical OTS devices for preliminary electrical evaluation, supported by material analyses and by AIMD simulations. Please find in the following some comments/remarks to be addressed to improve the paper quality.

Dear Reviewer 1:

We are very grateful to Reviewer 1 for his interest in the theme of our manuscript. We have performed many additional experiments and added the results to answer your questions, which are as follows.

As a general comment, what is missing at the end is a clear guideline to optimize the As content in GeAsS system. Is coming from simulations? Or from devices results? There are some trends, but not clearly correlated one to the other. We can conclude that As is responsible for a "reduced mobility" due to Ge-As and As-S formation, however it could form As clusters: it could be detrimental? Moreover, the paper title should be revised and focused on the real objective of the work: As content effects in $\text{GeS}_x\text{As}_{1-x}$ system.

Reply: Our manuscript mainly focuses on the experimental device performances, including ON/OFF current, endurance and operation voltage drift, etc., to optimize the As content in GeAsS system. The simulations were only used to find the inherent mechanisms for the performance changes with As content rather than optimizing the component. It can be seen from below that GeSAs_{25} is the one with the best performance among the four components.

Material	Feature Size (nm)	Thickness (nm)	Selectivity	J_{on} (MA/cm ²)	I_{off} (A)	V_{th} (V)	V_{h} (V)	Speed (ns)	Endurance	Thermal Stability
GeSAs_{20}	60	10	10^5	21	2.3×10^{-9}	3.3	~1.9	~10	10^7	450°C/30 min
GeSAs_{25}	60	10	10^4	16	1.2×10^{-8}	2.5	~1.5	~10	10^9	450°C/30 min
GeSAs_{43}	60	10	10^4	12	1.5×10^{-8}	2	~1.4	~10	~ 10^{10}	450°C/30 min
NGeCTe^{35}	32	15	10^4	12	~ 2×10^{-8}	~1.5	~1.2	—	—	400°C/30 min
AsTeGeSiN^{30}	30	40	10^3	11	~ 2×10^{-7}	~1.5	—	—	—	500°C/15 min
GeSe-based OTS ³²	50	5	10^3	—	10^{-6}	~1.4	~0.5	2	—	350°C/4 min
TeAsGeSiSe^{33}	350	20	10^4	0.44	5×10^{-9}	3	—	—	—	350°C/30 min
Ge-Se-Sb-N ³⁴	350	—	10^6	0.2	10^{-10}	2.5	~1.3	—	10^8	400°C/30 min
C-Te ³¹	30	~10	10^5	11	5×10^{-9}	~0.6	~0.3	<10	10^6	450°C/30 min

Table R1 Summary of ovonic threshold switching device performances after annealing using different materials. The properties highlighted in red indicate that they are superior to those of other materials, while the green ones refer to unsatisfactory performances of the material.

Fig. R1 Radar charts of different materials. The larger the area of the figure, the closer it is to the regular hexagon, indicating that the properties of the material are superior all around. Endurances of annealed NGeCTe, AsTeGeSiN, GeSe-based OTS, TeAsGeSiSe devices and the J_{on} of GeSe-based OTS are evaluated based on their as-deposited performances.

Table R1 is a summary of the device performances of annealed OTS devices reported in the literature, and the comparisons among different materials are shown as radar charts in **Fig. R1**. As we can see from **Table R1**, AsTeGeSiN device still operates normally after annealing at 500 °C for 15 minutes, which is the highest heat-treatment temperature in the table¹. However, I_{off} of the annealed device is only 0.2 μ A, which leads to a rather low storage density^{1,2}. Similarly, the I_{off} of the GeSe-based OTS material is only 1 μ A and its on/off ratio is 10^3 , which is the lowest among these materials³. Compared to GeSe, the leakage current of TeAsGeSiSe is 5 nA, but J_{on} drops to 0.44 MA/cm²⁴. Similar to TeAsGeSiSe, the on-state current density of annealed Ge-Se-Sb-N device is only 0.2 MA/cm², although its I_{off} is as low as 0.1 nA and the selective ratio is as high as 10^6 ⁵. CTe combines a J_{on} of 11 MA/cm² and a nA-scale I_{off} , while the device endurance decays from 10^8 to 10^6 cycles². In fact, 3D PCM requires comprehensive performance of OTS materials, so we visualized the performance of these devices from five perspectives: thermal stability, endurance, J_{on} , I_{off} , and selectivity as shown in **Fig. R1**. Evidently, I_{off} of annealed NGeCTe, GeSAs₂₅, and GeSAs₄₃ devices are relatively low, and exhibit high J_{on} without sacrificing the device endurance. However, GeSAs₂₅ and GeSAs₄₃ devices deliver larger J_{on} and lower I_{off} with relatively high lifetime, revealing higher competitive than NGeCTe⁶. However, a higher As content will lead to a decrease in V_{th} , which almost overlaps with V_h and squeezes the read margin, which is not what we hope to see. Clearly, As segregation occurs in the expired device through the EDS mappings after the endurance test as shown in rectangles of **Fig. R2**, and the higher the As content, the more obvious the phase segregation will be. That is to say, the concentration of As is not the higher the better. In conclusion, we believe that GeSAs₂₅ is the optimal component. The relevant description has been added to the manuscript.

Finally, based on your comments, we decide to change the title of the manuscript to “The Role of Arsenic in the Operation of Sulfur-based Electrical Threshold Switches”.

Fig. R2 Elemental mappings of fail device after endurance test. Arsenic in the rectangular region is segregated under 10^8 pulse operations

1) Line 30: "... incorporation of 20 at. % As brings a more than 100 °C increase in crystallization temperature...". This could be not valid for all the alloys. It would be better to specify since the abstract the alloy investigated.

Reply: Thanks for your comments and I have revised it in the manuscript. The result of the modification in Line 30 is shown below.

"We discovered that incorporation of As into GeS brings a more than 100 °C increase in crystallization temperature, remarkably improving the switching repeatability and prolonging the device lifetime."

2) Line 42: "These findings allow the use of precise control of the As concentration in OTS materials for improved-performance high-density 3D PCM applications". Again, it would be better to restrict to the alloy investigated: GeS+As.

Reply: Thanks for your comments and I have revised them in the manuscript. The result of the modification is shown below.

"These findings allow the precise performance control of GeSAs-based OTS materials for improved-performance high-density 3D PCM applications."

3) I think the title is too much generic, while the authors refer to a specific alloy investigation. As a reader, I would prefer to find better specified in the title the addressed system of the work, since the As effects strongly depends on its arrangement in the system and its bonds with the other elements.

Reply: Thanks for your comments and the title has been changed to "The Role of Arsenic in the Operation of Sulfur-based Electrical Threshold Switches."

4) The quality of the picture is not suitable for publication. Please, improve the pictures quality.

Reply: Thanks for your comments and I have improved the pictures quality.

5) In Fig. S7 the authors highlight a peak for GeS at 30°. Such peak is not indexed in standard databases for GeS. Which is the reference used? It is quite difficult to support by XRD results the improved stability by As introduction. Are the y-axis using the same scale?

Reply: Thanks for your comments. The corresponding peak at 30° is shown in **Fig. R3** which is captured from Materials Project (https://materialsproject.org/materials/mp-2242?chemsys=Ge-S#crystal_structure). And the separated XRD pictures of all four compositions are shown in **Figs. R4-R7**. The y-axes of GeS are different because of the presence of Si peak and the y-axes of other three components are in the same scale. In fact, XRD is a very simple and effective characterization method to demonstrate the improvement of thermal stability after the addition of As. We enable to determine the crystallization temperature of thin films with different components by carrying out XRD on different samples of the same component at different annealing temperatures. If the intensity of the scattered X-ray in the XRD results changes smoothly, it indicates that the material is still amorphous, and its crystallization temperature is higher than the annealing temperature at this time. If a peak with a clear half width appears in XRD, it indicates that crystallization has already occurred inside the material, and the crystallization temperature of the material is lower than the annealing temperature of the thin film. The only difficulty of this experiment lies in how to protect the film from vaporization during high-temperature annealing. GeSAs films can withstand an ascending annealing temperature without dissipation until 500 °C. However, when the annealing time exceeds 10 minutes at 500 °C, the dissipation cannot be prevented. That is why we did not obtain the specific crystallization temperatures of GeSAs₂₀, GeSAs₂₅, and GeSAs₄₃ through XRD.

Fig. R3 XRD of crystal GeS

Fig. R4 XRD of GeS annealed at different temperatures

Fig. R5 XRD of GeSAs₂₀ annealed at different temperatures

Fig. R6 XRD of GeSAs₂₅ annealed at different temperatures

Fig. R7 XRD of GeSAs₄₃ annealed at different temperatures

6) In Fig. S8 the ON current is defined as the current induced in the device at the threshold event. Such parameter is not of real interest since it depends on the threshold voltage and on the ON

resistance of the device. The ON resistance is related to the series external resistance only? If yes it should be specified that the graphs report the IV characteristics of the device in series with some external resistance.

Reply: Thank you for your comment. We have added some experiments to study the I - V performance using different series resistors which will greatly affect the ON resistance. As shown in **Fig. R8**, the circuit is used for the I - V test. Firstly, we change R1 to a 3.3 k Ω one and apply 50 consecutive triangular pulses to the device with different As contents. The responses are shown in **Fig. R9**, where devices with all four components operate normally, and it can also be observed that the consistency of device performance has improved after the addition of As. The performance distribution of different devices is shown in **Fig. R10**. Similarly, we also conduct the same test using a resistance of 5.5 k Ω , as shown in **Fig. R11** and **R12**. Clearly, the impact of RC is exacerbated as the resistance further increases. After summarizing the device performances of each component measured under different R1, it seems that the resistance of R1 has little effect on the V_{fire} , but leads to the slightly increase of V_{th} and V_{h} of the device with the same composition in **Fig. R13a**. This is because the ascending value of R1 shares higher voltage on it. Therefore, the OTS device and R1 require a higher applied voltage to turn on and the device also turns off at a higher V_{h} , which are consistent with the conclusion in the literature⁷. **Fig. R13b** demonstrates that the I_{on} of the device significantly influenced by R1 and the intrinsic ON resistance of GeSAs devices shown in **Fig. R14** are not affected by As concentration and R1 value. Besides, the changing trend of I_{on} with As concentration is consistent with V_{th} under the same R1, but I_{on} of the device with the same As content gradually decreases with the increase of R1, which is consistent with the conclusions in the literature, too³.

Fig. R8 The circuit of I-V test

Fig. R9 The responses of different GeSAs devices to 50 consecutive triangular pulses using 3.3 k Ω resistor

Fig. R10 The performance distribution of different GeSAs devices using 3.3 kΩ resistor

Fig. R11 The responses of different GeSAs devices to 50 consecutive triangular pulses using 5.1 kΩ resistor

Fig. R12 The performance distribution of different GeSAs devices using 5.1 kΩ resistor

Fig. R13 The variation of device performance with As concentration. **a.** The tendency of V_{fire} , V_{th} and V_h with the incorporation of As under different R1. Blue, green and red lines represent R1 of 1.2 k Ω , 3.3 k Ω and 5.1 k Ω , respectively. **b.** The change trend of I_{on} vs. As content using different R1.

Fig. R14 ON resistance of different GeSAs devices with different external resistors

7) In Fig. S9 in the graph the fail for GeS is reported to be at 400°C, while in the caption the authors specify an annealing at 450°C 30min. Which is the real thermal budget applied?

Reply: Thank you for pointing out the mistake. GeS film was annealed at 400 °C rather than 450 °C. We have modified the captions and provided a more detailed description of the thermal budget. The results of the modifications are as follows.

“ V_{fire} , V_{th} and V_h distributions of annealed GeSAs devices. GeS fails after 400 °C annealing and other GeSAs devices operate normally after 450 °C for 30 minutes.”

8) Line 140-150: the ON current calculation seems more dependent on the external resistance used than from the material local parameters (i.e. On current equal to V_{th} over ON resistance). It would be better to clarify if the author is referring more to the V_{th} of the material than to the ON current.

Reply: Thank you for your comment and I_{on} depends more on the external resistance indeed. In Line 140-150, we discuss the variation of I_{on} with As content, for the current density is a key to drive PCM. Here, we not only emphasize this point but also make a statement to prevent readers from misunderstandings because the I_{on} of the GeS device in this manuscript is inconsistent with previous work⁸, where the value of the I_{on} reaches 10 mA scale due to the W/GeS/TiN/Al device structure from bottom to top. Once a pulse is applied, an Al fuse will be formed across the TiN layer resulting in a lower ON resistance and there is no resistors to constraint the current. In this manuscript, the top electrode device is 40-nm TiN, and I_{on} depends on TiN and the series resistance, and the results are in line with another literature with the same structure⁹.

9) Line 145: since V_{th} could vary dependently on ramp rate used, a variation between AC and DC protocols is expected. Data in S2 etc. are obtained from DC or AC protocol? It would be better to specify it each time to avoid misunderstanding, and explain why this choice in the experimental description.

Reply: Thank you for your comments and we have made more specific descriptions of the S2 and other figures. The revised results are shown below.

In Fig. S1:

“...normal distributions. V_{fire} of GeS, GeSAs₂₅ and GeSAs₄₃ devices are obtained by the device responses to the 6 V triangular pulses, and V_{fire} of GeSAs₂₀ is measured by a 6.5 V one. V_{th} , V_h and I_{on} of devices with different As concentrations are determined by 3 V, 4.5 V, 4 V and 4 V pulses, respectively. The rising and falling edges of all pulses are 1 μ s. Moreover, I_{off} is measured through DC test and the step is 0.1 V. The V_{fire} values for...”

In Fig. S2:

“...threshold switching. V_{fire} of GeS, GeSAs₂₅ and GeSAs₄₃ devices are obtained by the device responses to the 6 V triangular pulses, and V_{fire} of GeSAs₂₀ is measured by a 6.5 V one. V_{th} , V_h and I_{on} of devices with different As concentration are determined by 3 V, 4.5 V, 4 V and 4 V pulses, respectively. The rising and falling edges of all pulses are 1 μ s. Moreover, I_{off} is measured through DC test and the step is 0.1 V. I_{off} refers to...”

In Fig. S5:

“...GeSAs₂₀ devices. 2.5 V and 3.5 V square pulses with 20 ns rising and falling edges, 100 ns pulse width and 50 ns interval were used in the endurance measurements, respectively. The dynamical...”

In Fig. S6:

“...devices. 3.5 V square pulses with 20 ns rising and falling edges, 100 ns pulse width and 50 ns interval were used in the endurance measurements. GeSAs₂₅...”

In Fig. 2b:

“...pulses. 100 consecutive triangular pulses with the amplitudes of 4 V, 4.5 V, 4 V and 4 V are applied to GeS, GeSAs₂₀, GeSAs₂₅ and GeSAs₄₃ annealed devices, respectively. The GeS...”

In Fig. S9:

“...30 min. The testing pulses are the same as 200-nm as-deposited GeSAs devices. The devices...”

In Fig. S10:

“...containing As. GeS devices failed after 400 °C annealing. V_{fire} of GeSAs₂₀, GeSAs₂₅ and GeSAs₄₃ devices are obtained by the device responses to the 8.5 V, 7 V and 7 V triangular pulses. V_{th} , V_h and I_{on} of devices with different As concentrations are determined by 5 V, 4 V and 4 V pulses, respectively. The rising and falling edges of all pulses are 1 μ s. Moreover, I_{off} is measured through DC test and the step is 0.1 V. In 20 at. % As...”

In Fig. S11:

“...OTS layers. V_{fire} of 5-nm, 10-nm and 20-nm GeS devices are obtained by the device responses to the 4.5 V, 6 V and 8 V triangular pulses. V_{th} , V_h and I_{on} of devices with different As concentrations are determined by 2.5 V, 3 V and 4 V pulses, respectively. The rising and falling edges of all pulses are 1 μ s. Moreover, I_{off} is measured through DC test and the step is 0.1 V. In 5 nm-GeS...”

In Fig. S12:

“...OTS layers. V_{fire} of 5-nm, 10-nm and 20-nm GeSAs₂₀ as-deposited devices are obtained by the device responses to the 5 V, 6.5 V and 10 V triangular pulses. V_{th} , V_h and I_{on} of devices with different As concentrations are determined by 3 V, 4.5 V and 7 V pulses, respectively. Besides, V_{fire} of 5-nm,

10-nm and 20-nm GeSAs₂₀ annealed devices are obtained by the device responses to the 5 V, 8.5 V and 10 V triangular pulses. V_{th} , V_h and I_{on} of devices with different As concentrations are determined by 3 V, 5 V and 7 V pulses, respectively. The rising and falling edges of all pulses are 1 μ s. Moreover, I_{off} is measured through DC test and the step is 0.1 V. In 5-nm...

In Fig. S13:

“...OTS layers. V_{fire} of 5-nm, 10-nm and 20-nm GeSAs₂₅ as-deposited devices are obtained by the device responses to the 5 V, 6 V and 9 V triangular pulses. V_{th} , V_h and I_{on} of devices with different As concentrations are determined by 3 V, 4 V and 5.5 V pulses, respectively. Besides, V_{fire} of 5-nm, 10-nm and 20-nm GeSAs₂₅ annealed devices are obtained by the device responses to the 5 V, 7 V and 9 V triangular pulses. V_{th} , V_h and I_{on} of devices with different As concentrations are determined by 3 V, 4 V and 5 V pulses, respectively. The rising and falling edges of all pulses are 1 μ s. Moreover, I_{off} is measured through DC test and the step is 0.1 V. In 5-nm...”

In Fig. S14:

“...OTS layers. V_{fire} of 5-nm, 10-nm and 20-nm GeSAs₄₃ as-deposited devices are obtained by the device responses to the 5 V, 6 V and 9 V triangular pulses. V_{th} , V_h and I_{on} of devices with different As concentrations are determined by 3 V, 4 V and 5.5 V pulses, respectively. Besides, V_{fire} of 5-nm, 10-nm and 20-nm GeSAs₄₃ annealed devices are obtained by the device responses to the 5 V, 7 V and 9 V triangular pulses. V_{th} , V_h and I_{on} of devices with different As concentrations are determined by 3 V, 4 V and 5 V pulses, respectively. The rising and falling edges of all pulses are 1 μ s. Moreover, I_{off} is measured through DC test and the step is 0.1 V. In 5-nm...”

In Fig. S15:

“...30 min. 4 V square pulses with 20 ns rising and falling edges, 20 ns pulse width and 50 ns interval were used in the endurance measurements. The device...”

In Fig. S16:

“...30 min. 4 V square pulses with 20 ns rising and falling edges, 20 ns pulse width and 50 ns interval were used in the endurance measurements. The device...”

We measure the V_{fire} , V_{th} , V_h and I_{on} of the devices by applying pulses, while I_{off} is obtained from DC protocol. The reason for this is due to using pulses is more relevant to the real application scenario than DC and the device cannot be repeated operating through DC operations because the width of the read pulse of DC test is 3 ms which may cause significant damage to the device.

10) Line 156: how the authors take into account the RC times? The ON resistance of the device is not calculated. How much does it affect the switching on/off speed? Since no trend is observed in S4d it would be better either to remove this calculation or to better support the result with the parasitic analysis of the used equipment.

Reply: Thanks for your comment and here are the efforts we have made in reducing the RC impacts during the speed testing. The speed test circuit is just like I - V test in Fig. R8, and RC delay comes not only from the charge/discharge of the device during the switching, but also from the wiring of the circuit itself where inter-pole and coupling capacitances are inevitable, so it is hard to estimate the extent of RC effect. But we still find ways to alleviate the delay by decreasing the resistance of R1, R2 and integrating R1 with the circuit, which not only reduce the capacitance impact between the oscilloscope and R2, but also effectively eases the current overshoot. Under these two methods, we successfully boost the speed from the hundreds of nanoseconds to less than ten nanoseconds. As for ON resistance, the resistance of on-state device is shown in Fig. R14, while the distribution of

ON/OFF speed using different R1 is shown in **Fig. R15**. With the increasing of R1, the ON resistance of GeSAs devices with different As contents fluctuates within the range of 800-3000 Ω , indicating that the content of As and the external resistance do not affect the intrinsic ON resistance of GeSAs. Besides, as the external resistance increases, the influence of capacitance effect increases, which also leads to an increase in the deviation of speed, but ON resistance of the device has almost no effect on the operating speed of the device. The ON-speed ranges are between 8 and 15 ns and the OFF speeds range from 9 to 15 ns. However, this is not the reason to remove this section, because speed is a key parameter of memory, and RC delay only increases the ON/OFF time. Our measured results have shown that GeSAs devices possess a nanosecond level of ON/OFF speed. Although our testing still has flaws, it cannot be deleted.

Fig. R14 ON resistance of different GeSAs devices with different external resistors

Fig. R15 ON/OFF speed of different GeSAs devices with different external resistors

11) Data from Fig.1e cannot be read. Moreover, there is no statistics. Is there a real trend increasing As content? Is this trend confirmed in cycled (and after how many cycles?) devices?

Reply: Thank you for your comment. We conduct statistical analysis on the results of V_{th} drift, as shown in **Fig. R16a** and replace **Fig. 1e** with it. As for the test method, we take GeS as an example. Firstly, we fire the device by a 6 V triangular pulse, and it is followed by a 3 V one after 1 μ s and

we take this point as the zero point of time. Then, we apply the same pulse after 1 minute, 1 hour, and 1 day and record the V_{th} of the device each time. The triangular pulses used to initiate the GeSAs₂₀, GeSAs₂₅, and GeSAs₄₃ devices are all 6 V, while the voltage amplitudes of the V_{th} sampling are 4.5 V, 4 V and 4 V, respectively. All the rising and falling edges of triangular pulses are 1 μ s and ten devices of each composition are tested. From Fig. R16a, we observe that the average V_{th} of GeS devices drift from 1.46 V to 1.72 V after 1 day, with a change rate of 18%. After doping with As, V_{th} drifts from 2.98 V to 3.16 V, with a change rate of 6%. When the As content increases to 25 at.%, V_{th} increases from 2.64 V to 2.75 V, with a change rate of 4%. In the GeSAs₄₃ devices, V_{th} climbs from 2.23 V to 2.64 V, while the rate of change increases to 18%, which is consistent with the conclusion that V_{th} spontaneously increase over time, and the addition of As can effectively suppress the drift. Afterwards, we apply 1000 continuous triangular pulses which is the same as the sampling pulse of each component to all GeSAs device after the fire process. V_{th} drift tests are carried out on the device after 1000-time operations using the same method. It can also be observed that V_{th} gradually increases with the extension of interval in Fig. R16b. When the As content is 20 at.%, the increase in V_{th} after 1 day is 460 mV. As the concentration of As continues to increase, the increase in V_{th} drops to 440 mV. In GeSAs₄₃ devices, V_{th} drifts 700 mV after 1 day, which is consistent with the conclusion obtained in Fig. R16a, 25 at.% As incorporation can effectively suppress V_{th} drift even in cycled devices.

Fig. R16 a. V_{th} drift of as-deposited GeSAs devices with time. **b.** V_{th} drift of GeSAs devices after 1000-time operations.

12) The reported endurance tests are interesting. However, what it is missing is a statistic to validate the observed trend.

Reply: Thank you for your comments and we further carry on endurance tests on several as-deposited and annealed devices as shown in Fig. R17-R18. Endurance of as-deposited GeS device is in line with the result reported in the literature⁸.

Fig. R17 Endurance tests of multiple GeSAs devices

Fig. R18 Endurance tests of multiple GeSAs annealed devices

13) In the endurance plots, we can observe a clear evolution of the ON resistance of the device along cycling, and even if bottom right IV plots of S14 and S15 are not perfectly clear, they seem to reveal an evolution of the material along cycling. How the atomic migration/composition evolution in the formed filament/channel can be excluded (line 275)? A TEM image after the endurance test would have been interesting to validate such hypothesis, since in literature several proofs of phase separation in such system have been previously reported.

Reply: Thank you for these valuable comments and we have updated all the figures. We really agree with what you said. The atomic migration/composition evolution were not excluded. We have conducted further research on the causes of changes in I_{off} , and here are the TEM images of GeSAs₄₃ device after operating by 10^4 square pulses with the amplitude of 4 V, 20-ns rising and falling edges, 20-ns pulse width, and 50-ns interval. From **Fig. R19**, the device structure remains intact, and it can be seen that there is an uneven distribution of elements in the red rectangular area on the right side

of the material layer. In order to further study the distribution of elements in this area, we conduct an EDS line scanning on the position of the white line in the TEM image, as shown in **Fig. R20**. We can observe that not only does the aggregation of As occur in the material layer, but also the diffusion of C appears. The pulse-operation results in a chaotic element distribution and there is an obvious phase separation phenomenon. Afterwards, we enlarge the left, middle, and right regions of the material layer and perform fast Fourier Transform (FFT) on each region, as shown in **Fig. R21-23**. The FFT pictures in all regions of the material layer exhibit diffuse diffraction phenomena with no lattice fringes, indicating that GeSAs₄₃ remains amorphous at this time but the formation of Ge-Ge filaments is not observed. In fact, the filaments are hard to be captured through TEM because the formation of conductive filaments in OTS materials is only a mild structural change locally, not an overall change reported in several literature.¹⁰⁻¹³ Nevertheless, this possibility still exists, **thus we assume that the increase in I_{off} may be jointly caused by phase separation, C-layer diffusion or the formation of Ge-Ge filaments. This statement has been added in the revised manuscript.**

Fig. R19 TEM image and EDS mappings of GeSAs₄₃ device operated for 10⁴ cycles

Fig. R20 Result of EDS line scanning at the position of the white line

Fig. R21 High resolution TEM image of the left region of the GeSAs layer

Fig. R22 High resolution TEM image of the middle region of the GeSAs layer

Fig. R23 High resolution TEM image of the right region of the GeSAs layer

14) Line 281: looking at the current densities implied in the switching mechanism it is reasonable to expect temperatures in the ON region of the device well higher than 500°C. What could be interesting is to validate a low crystallization speed thanks to "As" introduction, since the high crystallization temperature could hinder crystallization phenomena along the BEOL thermal budget of the fabrication, but its correlation with endurance is not direct. Maybe the authors could better explain such link.

Reply: Thank you for your comment. In terms of endurance, As content has an effect on the upward trend of I_{off} with the increase of pulse numbers. We calculate the standard deviation of the I_{off} under different amounts of testing pulses as 0.1 μ A, 68 nA, 20 μ A and 28 nA for four compositions, and we find that with the increase of As content, the fluctuation amplitude of I_{off} decreases with the increase of pulse counts which is shown in **Fig. R17**. The main reason for this fluctuation originates from crystallization, so an increase in As hinders the crystallization which improve the endurance. The function of As is more evident in annealed devices. When the As content is 20 at.%, the endurance of the annealed device is only 10^7 cycles. However, as the As content continues to increase, the lifetime prolongs to 10^9 cycles, and it further reaches 9×10^9 cycles in annealed GeSAs₄₃ device. This indicates that the addition of As has indeed played a role in inhibiting atomic migration at high temperatures, thereby improving device endurance.

15) Fig.3e should report either in the caption, either in the graph the indexation of the deconvoluted peaks.

Reply: Thank you for your comment and we have revised the relevant description of the Raman

results and report the indexation of the peaks in the figure.

The latest Raman results and the corresponding atomic groups for each peak are shown in **Fig. R24** and **Table R2**.

Figure R24 Raman results of GeSAs films

Wavenumber (cm ⁻¹)	Assignments
210,212,213,407	Ge-S chains
248	Nonmolecular As-As bonds
255,262	S ₃ Ge-GeS ₃ ethane-like unit
291	Ge-Ge bond vibrations in S ₃ Ge-GeS ₃
360,361,362	As-S vibrations in As ₄ S ₄
366	Ge-S bonds in ethane-like S ₃ Ge-GeS ₃ and edge-shared Ge ₂ S ₆ tetrahedra
397,400	Interactions between AsS ₃ pyramids

Table.R2 Raman peaks and corresponding atomic groups

16) Line 336: the Raman analysis evidencing As-As vibration seems not reported in the description.

Reply: Thank you for your comment and we have added the relevant description of the Raman results and references. The results are shown below.

“However, the domination of Ge-S bonds is replaced in GeSAs₄₃ where the number of As-As bonds emerge in abundance at 248 cm⁻¹ ⁵⁰, while other peaks at 210, 360 and 400 cm⁻¹ stand for Ge-S chains¹⁷, As₄S₄^{46,47} and AsS₃⁴⁷ atomic groups.”

17) Line 346: As-As clusters formation in Fig.3d and 3e is not clear/evidenced. Clusters formation can also be interpreted as a segregation phenomenon. How such clusters should evolve during the device operations?

Reply: Thank you for your comment. Raman spectroscopy is based on the GeSAs thin film samples deposited by PVD, and the interval between preparation and testing is short, making it unlikely that the thin film components will exhibit phase separation. In addition, from the ON resistance in **Fig. R14**, we can also qualitatively find that the As content does not affect the on-state of the device. Although it is difficult to observe As-As clusters experimentally, we can qualitatively evaluate the possibility of As separation or aggregation through the difference in calculated free energy. **Fig. R25** illustrated the four models with different degrees of As aggregation, in which the degrees of As separation shows: model a (total As aggregation) > model b > model c > model d (random As distribution), and **Table R3** displayed the total energy of model a-d, and the model (a) possesses the lowest total energy, model (d) possesses the largest total energy. Meanwhile, we calculated the energy difference per atom ($\Delta E/\text{atom}$) between each model and model (a), indicating that higher degree of As segregation leads lower $\Delta E/\text{atom}$. The comparison of energy and $\Delta E/\text{atom}$ suggested that the As atoms have a tendency to segregate after many device operations for a-GeSAs₄₃. **Figs. R19-R20** also show the segregation phenomenon for As clusters.

Fig. R25 a-GeSAs₄₃ models with various degrees of As-segregation. The model generated by **a.** artificially clustered As atoms. **b and c.** model a melted at 2000 K temperature for 1.5 ps and 6 ps, respectively. Longer melting time leads to more disordered configurations. **d.** randomly replaced Ge and S atoms by As atoms in a-GeS model. All the models were relaxed at 500 K for 3 ps and then performed structural optimization at 0 K to obtain total energy of each model.

Table.R3 Total energy of a-GeSAs₄₃ models with various degrees of As-segregation

Model	a	b	c	d
Energy (eV)	-1336.93	-1333.47	-1328.76	-1325.62
$\Delta E/\text{atom}$ (eV)	0	0.0115	0.0272	0.0377

Fig. R26 Evolution of the structure and the corresponding DOS for a-GeSAs₄₃ **a.** eigen state. **b.** hole excited state. The location of traps changed but the atomic clusters associated with the trap states changed little, only the more bond aligned Ge-centered and As-centered defective octahedrons.

18) Raman spectra were performed on as-dep layers: it would have been interesting to perform same analyses on annealed samples, to observe the structural evolution at high temperature (not visible in XRD experiments).

Reply: Thank you for your comment and we fully agree with your viewpoint. Therefore, we decide to anneal GeSAs films of different components first and carry out Raman tests on them. To prevent material evaporation, a protective layer is needed, while SiO₂ used in XRD shows Raman peaks in 100-500 cm⁻¹ ¹⁴. Instead, we deposit a layer of C which shows no Raman peaks in 100-500 cm⁻¹, and its crystallization temperature is higher than 600 °C. Afterwards, GeS film is annealed at 400 °C for 10 minutes, while GeSAs₂₀, GeSAs₂₅ and GeSAs₄₃ films are annealed at 450 °C for 30 minutes. Then Raman spectra are obtained by LabRAM HR800 Raman spectrometer and the results before and after annealing are shown in **Fig. R27**. It can be observed that there is almost no change in the peak positions of GeSAs₂₀, GeSAs₂₅, and GeSAs₄₃ after annealing in **Fig. R27b**. However, GeS shows significant changes after annealing with peaks at 210 and 249 cm⁻¹ representing the B_{2g}

vibration mode¹⁵ and bond bonding motions of S atoms in GeS₄ tetrahedra¹⁶ which appear in crystal phase. Also, GeS still contains amorphous part at this time with peak at 270 cm⁻¹ corresponding to SGe₃-S_{6/3} groups¹⁷, peak at 288 cm⁻¹ corresponding to SGe₃ pyramids¹⁷, peak at 350 cm⁻¹ corresponding to edge-shared GeS₄¹⁷, and peak at 388 cm⁻¹ corresponding to S₃Ge-GeS₃ units¹⁷. Hence, we conclude that GeS crystallizes at 400 °C, and GeSAs materials enable to withstand thermal shock at 450 °C, which is consistent with XRD. We have added this part to the supplementary materials.

Fig. R27 Raman of GeSAs films a. Raman spectra of as-deposited GeSAs films **b.** Raman spectra of annealed GeSAs films. GeS film is annealed at 400 °C for 10 minutes and others are annealed under 450 °C for 30 minutes.

The subject of the present manuscript is a detailed study the Ovonic Threshold Switching (OTS,) of the GeSAs chalcogenide glass. The OTS behaviour corresponds to a volatile electronic commutation between highly and low resistive state when a voltage higher than the threshold voltage is applied. OTS materials are studied to replace silicon-based traditional selector technologies to enable the stacking of the memory with the selector with equivalent feature sizes, increasing indeed the density of the memory. The main challenge concerning OTS development is to find optimal compositions that allows the material to stay amorphous, when operating, but also when the thermal budget reaches above 400 °C during 3 hours. The manuscript is well-written. The results are convincing, but for the exceptions noted below, and the insight provided will certainly prove to be of great interest well beyond the phase change material community.

Dear Reviewer 2:

Many thanks to Reviewer 2 for the high evaluation of our work. We are glad that you can identify with our achievements. The following is our response to your comment, hoping to satisfy all your concerns. We sincerely hope that you can agree to publish our manuscript.

1. OTS materials by their engineering are hard to crystallize, however conductive and stables crystallites can gradually form within the layer and lead tdevice failure in endurance. This is usually observed as a progressive decreasing of the OFF resistance. The authors presented in Fig.2 some nice TEM images from which resolution it is not possible to exclude the presence of nano crystals after annealing. Are the authors able to provide higher resolution measurements?

Reply: Thank you for your comment. The higher resolution TEM image is shown in **Figs. R28-R30**. We study the crystallization situation in various regions of the GeSAs layer using fast Fourier transform images. We can see that the FFT images all exhibit diffuse diffraction phenomena without any bright sparkles, indicating that the GeSAs material layer remains amorphous after annealing.

Fig.R28 Left side of high resolution of annealed GeSAs₄₃ device

Fig. R29 Middle side of high resolution of annealed GeSAs₄₃ device

Fig. R30 Right side of high resolution of annealed GeSAs₄₃ device

2. The drift phenomenon (aging) is linked to the amorphous relaxation of the material and it is often attributed, both in PCM and OTS, to the Ge-Ge bonds presence (to the fraction of tetrahedrally coordinated Ge). For the present GeSAs alloys the authors point at the formation of Ge-Ge filaments. I would be interested to know the number of atoms corresponding to an average sized filament. Would it be possible to directly observe it by TEM measurements?

Reply: Thanks for your valuable comment. In fact, Ge-Ge filament theory is mentioned in many references¹⁰⁻¹³, in which Ge-Ge filament was reported to consist of only several or dozens of atoms. Therefore, this Ge-Ge filament is very hard to be observed by the TEM technology, as shown in **Fig. R21-R23**, taken from an GeSAs₄₃ device after 10⁴-time operations. Further investigations with advanced technology are needed. In spite of this, the Ge-Ge filament cannot be excluded. We used “may be due to the formation of Ge-Ge conductive filaments” in our statement.

Fig. R21 High resolution TEM image of the left region of the GeSAs layer

Fig. R22 High resolution TEM image of the middle region of the GeSAs layer

Fig. R23 High resolution TEM image of the right region of the GeSAs layer

3. The global composition of OTS is often complex in order to fulfil the specifications: high thermal stability, low leakage current, high endurance and tunable threshold voltage. To further help the reader to judge the novelty of the manuscript the performance comparison that is presented in FigS16 should not only mentioned in the main text, but better and more extensively summarized. Although measurement protocol pulse (applied and measured pulses) can be extrapolated from Fig1b and are explained in the caption of FigS4, it would be useful if the authors could write in Methods to look for protocols in supplementary material.

Reply: Thank you for your comment. We have added the performance comparison of annealed devices, as shown in the **Table 1** and **Fig. 3**, in the revised manuscript.

Table .1 Summary of ovonic threshold switching device performances after annealing using different materials										
Material	Feature Size (nm)	Thickness (nm)	Selectivity	J_{on} (MA/cm ²)	I_{off} (A)	V_{th} (V)	V_h (V)	Speed (ns)	Endurance	Thermal Stability
GeSAs ₂₀	60	10	10 ⁵	21	2.3×10 ⁻⁹	3.3	~1.9	~10	10 ⁷	450°C/30 min
GeSAs ₂₅	60	10	10 ⁴	16	1.2×10 ⁻⁸	2.5	~1.5	~10	10 ⁹	450°C/30 min
GeSAs ₄₃	60	10	10 ⁴	12	1.5×10 ⁻⁸	2	~1.4	~10	~10 ¹⁰	450°C/30 min
NGeCTe ³⁵	32	15	10 ⁴	12	~2×10 ⁻⁸	~1.5	~1.2	—	—	400°C/30 min
AsTeGeSiN ³⁰	30	40	10 ³	11	~2×10 ⁻⁷	~1.5	—	—	—	500°C/15 min
GeSe-based OTS ³²	50	5	10 ³	—	10 ⁻⁶	~1.4	~0.5	2	—	350°C/4 min
TeAsGeSiSe ³³	350	20	10 ⁴	0.44	5×10 ⁻⁹	3	—	—	—	350°C/30 min
Ge-Se-Sb-N ³⁴	350	—	10 ⁶	0.2	10 ⁻¹⁰	2.5	~1.3	—	10 ⁸	400°C/30 min
C-Te ³¹	30	~10	10 ⁵	11	5×10 ⁻⁹	~0.6	~0.3	<10	10 ⁶	450°C/30 min

Table.1 Summary of ovonic threshold switching device performances after annealing using different materials. The properties highlighted in red indicate that they are superior to those of other materials, while the green ones refer to unsatisfactory performances.

Fig. 3 Radar charts of different materials. The larger the area of the figure, the closer it is to the regular hexagon, indicating that the properties of the material are superior all around. Endurances of annealed NGeCTe, AsTeGeSiN, GeSe-based OTS, TeAsGeSiSe devices and the J_{on} of GeSe-based OTS are evaluated based on their as-deposited performances.

We also added the protocols of electrical performance shown in **Fig. 1** and **Fig. 2** in the Method. In the supplementary materials, the protocols for the supplementary material data have been supplemented in the caption of every picture. The revised results are as follows.

Performance comparison part:

“...Compared with reported OTSs, annealed GeSAs devices present a better overall performance, as shown in Table. 1. As we can see from the table, AsTeGeSiN device still operates normally after annealing at 500 °C for 15 minutes, which is the highest heat-treatment temperature³⁰. However, I_{off} of the annealed device is only 0.2 μA , which leads to a rather low storage density^{30,31}. Similarly, the I_{off} of the GeSe-based OTS material is only 1 μA and its on/off ratio is 10^3 , which is the lowest among these materials³². Compared to GeSe, the leakage current of TeAsGeSiSe is 5 nA, but J_{on} drops to 0.44 MA/cm², that is insufficient to drive the PCM³³. Similar to TeAsGeSiSe, the on-state current density of annealed Ge-Se-Sb-N device is only 0.2 MA/cm², although its I_{off} is as low as 0.1 nA and the selective ratio is as high as 10^6 ³⁴. CTe combines a J_{on} of 11 MA/cm² and a nA-scale I_{off} , while the device endurance decays from 10^8 to 10^6 cycles³¹. In fact, 3D PCM requires comprehensive performance of OTS materials, so we visualize the performance of these materials from five perspectives: thermal stability, endurance, J_{on} , I_{off} , and selectivity as shown in Fig. 3. Evidently, I_{off} of annealed NGeCTe, GeSAs₂₅, and GeSAs₄₃ devices are relatively low, and exhibit high J_{on} without sacrificing the device endurance. However, GeSAs₂₅ and GeSAs₄₃ devices deliver larger J_{on} and lower I_{off} with relatively high lifetime, revealing higher competitive than NGeCTe³⁵. However, a higher As content will lead to a decrease in V_{th} , which almost overlaps with V_{h} and squeezes the read margin, which is not what we hope to see.

Besides, the experimental results demonstrate that moderate As incorporation could significantly reduce the leakage current and suppress the V_{th} drift, and, most importantly, it strongly enhances the thermal stability of OTS materials, improving the switching repeatability and prolonging the device lifetime, therefore enabling a processing-line-compatible OTS selector with superior properties for 3D memory applications. In conclusion, we believe that GeSAs₂₅ is the optimal component.

Protocol part:

Methods

Device preparation and measurement. The 10 nm-GeSAs layers of all components were RF-sputtered, utilizing GeS, (GeS)₈₀As₂₀, (GeS)₇₅As₂₅ and (GeS)₅₇As₄₃ alloy targets using a power of 25 W. The 5nm C layers and top TiN electrodes were deposited by DC-sputtering using powers of 40 and 75 W, respectively. The device performances were characterized by a Keithley 4200A-SCS instrument. Fig. 1c was obtained by applying 100 consecutive 3 V, 4 V, 3.5 V and 3.5 V triangular pulses to GeS, GeSAs₂₀, GeSAs₂₅ and GeSAs₄₃ as-deposited devices initialized by a 6.5 V triangular pulse for GeSAs₂₀ and a 6 V one for other three. Similarly, the pulses used to fire the annealed devices in Fig. 2b are 8.7 V, 7 V and 7V for GeSAs₂₀, GeSAs₂₅ and GeSAs₄₃. The 100 pulses for operation are 5 V, 4 V and 4V, respectively. As for V_{th} drift, a combination of a high and a low triangular pulse with 1 μs interval was applied firstly to the device. The higher one is used to fire the device, while the lower one is used to measure the V_{th} and we took the moment that the lower pulse was input as zero point. The amplitudes of pulses for firing are 6 V, 6.5 V, 6 V and 6 V with the increasing of As concentration. And the testing pulses are 3 V, 4 V, 4V and 4 V. In Fig. 2d, as-deposited GeSAs₄₃ devices with the OTS layer thicknesses ranging from 5 to 20 nm are fired by a 5 V, 6 V and 9 V triangular pulse and operated by a 3 V, 4 V and 5.5 V one, respectively. A 5 V, 7 V and 9 V triangular pulse is used to fire the annealed devices from 5 to 20 nm thickness. V_{th} was

obtained by a 3 V, 4 V and 5 V pulse for each. All the rising and falling edge periods of the triangular pulse are 1 μ s. Protocols of detailed measurements in supplement material are mentioned in their captions.

Other protocols:

In Fig. S1:

“...normal distributions. V_{fire} of GeS, GeSAs₂₅ and GeSAs₄₃ devices are obtained by the device responses to the 6 V triangular pulses, and V_{fire} of GeSAs₂₀ is measured by a 6.5 V one. V_{th} , V_{h} and I_{on} of devices with different As concentrations are determined by 3 V, 4.5 V, 4 V and 4 V pulses, respectively. The rising and falling edges of all pulses are 1 μ s. Moreover, I_{off} is measured through DC test and the step is 0.1 V. The V_{fire} values for...”

In Fig. S2:

“...threshold switching. V_{fire} of GeS, GeSAs₂₅ and GeSAs₄₃ devices are obtained by the device responses to the 6 V triangular pulses, and V_{fire} of GeSAs₂₀ is measured by a 6.5 V one. V_{th} , V_{h} and I_{on} of devices with different As concentration are determined by 3 V, 4.5 V, 4 V and 4 V pulses, respectively. The rising and falling edges of all pulses are 1 μ s. Moreover, I_{off} is measured through DC test and the step is 0.1 V. I_{off} refers to...”

In Fig. S5:

“...GeSAs₂₀ devices. 2.5 V and 3.5 V square pulses with 20 ns rising and falling edges, 100 ns pulse width and 50 ns interval were used in the endurance measurements, respectively. The dynamical...”

In Fig. S6:

“...devices. 3.5 V square pulses with 20 ns rising and falling edges, 100 ns pulse width and 50 ns interval were used in the endurance measurements. GeSAs₂₅...”

In Fig. 2b:

“...pulses. 100 consecutive triangular pulses with the amplitudes of 4 V, 4.5 V, 4 V and 4 V are applied to GeS, GeSAs₂₀, GeSAs₂₅ and GeSAs₄₃ annealed devices, respectively. The GeS...”

In Fig. S9:

“...30 min. The testing pulses are the same as 200-nm as-deposited GeSAs devices. The devices...”

In Fig. S10:

“...containing As. GeS devices failed after 400 °C annealing. V_{fire} of GeSAs₂₀, GeSAs₂₅ and GeSAs₄₃ devices are obtained by the device responses to the 8.5 V, 7 V and 7 V triangular pulses. V_{th} , V_{h} and I_{on} of devices with different As concentrations are determined by 5 V, 4 V and 4 V pulses, respectively. The rising and falling edges of all pulses are 1 μ s. Moreover, I_{off} is measured through DC test and the step is 0.1 V. In 20 at. % As...”

In Fig. S11:

“...OTS layers. V_{fire} of 5-nm, 10-nm and 20-nm GeS devices are obtained by the device responses to the 4.5 V, 6 V and 8 V triangular pulses. V_{th} , V_{h} and I_{on} of devices with different As concentrations are determined by 2.5 V, 3 V and 4 V pulses, respectively. The rising and falling edges of all pulses are 1 μ s. Moreover, I_{off} is measured through DC test and the step is 0.1 V. In 5 nm-GeS...”

In Fig. S12:

“...OTS layers. V_{fire} of 5-nm, 10-nm and 20-nm GeSAs₂₀ as-deposited devices are obtained by the device responses to the 5 V, 6.5 V and 10 V triangular pulses. V_{th} , V_{h} and I_{on} of devices with different As concentrations are determined by 3 V, 4.5 V and 7 V pulses, respectively. Besides, V_{fire} of 5-nm, 10-nm and 20-nm GeSAs₂₀ annealed devices are obtained by the device responses to the 5 V, 8.5 V

and 10 V triangular pulses. V_{th} , V_h and I_{on} of devices with different As concentrations are determined by 3 V, 5 V and 7 V pulses, respectively. The rising and falling edges of all pulses are 1 μ s. Moreover, I_{off} is measured through DC test and the step is 0.1 V. In 5-nm...

In Fig. S13:

“...OTS layers. V_{fire} of 5-nm, 10-nm and 20-nm GeSAs₂₅ as-deposited devices are obtained by the device responses to the 5 V, 6 V and 9 V triangular pulses. V_{th} , V_h and I_{on} of devices with different As concentrations are determined by 3 V, 4 V and 5.5 V pulses, respectively. Besides, V_{fire} of 5-nm, 10-nm and 20-nm GeSAs₂₅ annealed devices are obtained by the device responses to the 5 V, 7 V and 9 V triangular pulses. V_{th} , V_h and I_{on} of devices with different As concentrations are determined by 3 V, 4 V and 5 V pulses, respectively. The rising and falling edges of all pulses are 1 μ s. Moreover, I_{off} is measured through DC test and the step is 0.1 V. In 5-nm...”

In Fig. S14:

“...OTS layers. V_{fire} of 5-nm, 10-nm and 20-nm GeSAs₄₃ as-deposited devices are obtained by the device responses to the 5 V, 6 V and 9 V triangular pulses. V_{th} , V_h and I_{on} of devices with different As concentrations are determined by 3 V, 4 V and 5.5 V pulses, respectively. Besides, V_{fire} of 5-nm, 10-nm and 20-nm GeSAs₄₃ annealed devices are obtained by the device responses to the 5 V, 7 V and 9 V triangular pulses. V_{th} , V_h and I_{on} of devices with different As concentrations are determined by 3 V, 4 V and 5 V pulses, respectively. The rising and falling edges of all pulses are 1 μ s. Moreover, I_{off} is measured through DC test and the step is 0.1 V. In 5-nm...”

In Fig. S15:

“...30 min. 4 V square pulses with 20 ns rising and falling edges, 20 ns pulse width and 50 ns interval were used in the endurance measurements. The device...”

In Fig. S16:

“...30 min. 4 V square pulses with 20 ns rising and falling edges, 20 ns pulse width and 50 ns interval were used in the endurance measurements. The device...”

Reviewer #3 (Remarks to the Author):

The authors present an interesting and detailed investigation (first-principles simulations, electrical and physical characterization) of As-containing ovonic threshold switching materials that are currently required for selector application. They show that with increasing As content in GeAsS, the thermal stability of the material increases and leads to improved selector properties: longer endurance, better selectivity and lower threshold voltage drift. By using first-principles simulations, coupled with Raman spectroscopy, they show that the improvements are due to As dual electron donor/acceptor property and high As-S bond strength, which makes the atomic diffusivity slower and the probability to form conductive Ge-Ge chains lowers with increasing As content. Electrical conductivity is motivated with Poole-Frenkel model and electronic properties (the mobility gap and the trap levels), that were extracted both with experimental and DFT techniques.

It is noteworthy to see how the increasing content of As atoms in the composition can explain the non-monotonic evolution of the threshold voltage, mobility gap and the related parameters. The presented understanding will enable the research community to apply the principles of electron-counting and bond-strength to continue the selector material fine-tuning and eventually come-up with an As-free OTS selector material. The conclusions are sound and supported by the theoretical and experimental findings, the theoretical methodology is state-of-the-art, which makes the work of

very high quality and worth publishing. However, the novelty of the work is probably compromised, to be considered for Nature Communications, since (part of) the study was reported before (<https://doi.org/10.1016/j.scriptamat.2022.114834>).

Reply: Thank you very much for your comments and we are very grateful for your high evaluation of our manuscript. As for the novelty, the article you mentioned mainly discusses that As improves the thermal stability of GeAsSe system by calculations. The conclusion has already been put forward before and reached a consensus¹⁸. Besides, the calculation section related to thermal stability is only a small part of our entire article, while the focus is still on experiments and the calculation only serves as an auxiliary explanation.

In fact, due to the limitations of process and equipment, there has always been a lack of research on the effect of As content on device performance. Therefore, we set off from the experimental aspect through adjusting the As content in GeS system and find that As reduces the I_{off} of the device, improves the performance consistency, suppresses the V_{th} drift, extends the device endurance to $\sim 10^{10}$ cycles and improves the thermal stability so that it can withstand a high temperature of 450 °C in the back-end-of-line (BEOL) process. Afterwards, we obtain the structure and energy band characteristics of different GeSAs materials through experimental characterization. X-ray diffraction and transmission electron microscopy confirm that the crystallization temperatures of GeSAs exceed 500 °C. Through Raman spectroscopy, it is found that AsS₃ pyramids are formed after As doping. When the concentration of As reaches 43 at.%, As-As homopolar bonds appear. The photothermal deflection spectrum shows that after As doping, the band gap widths and the densities of trap states firstly increase and then decrease, where the turning point is at 20 at.% As concentration corresponding to the change trend of I_{off} with As concentration. In GeSAs₄₃, a new trap state appears in the band gap of GeSAs, which may be related to the As-As bonds in line with the Raman results. Finally, combining the experimental results with *ab-initio* calculations, it is found that stronger As-S bonds are formed after the addition of As that hinder the atomic migration, thereby improving the thermal stability and suppressing the V_{th} drift. By calculating the density of states and IPR, the main structure of trap states in GeSAs is Ge-Ge bond/chain. As will participate in the formation of Ge-Ge trap states at low As content. However, when the As content increases to 20 at.%, Ge-Ge bond/chain trap states have already saturated, leading to an increase in trap state concentration first and then a decrease. When the As content reaches 43 at.%, trap states with As-As bond/chain as the main structure will be formed.

We sincerely hope my explanation would satisfy you and please reconsider allowing our manuscript to be published.

For additional manuscript improvement suggestions, please see the commented pdf file (attached) 1.

Missing Verb

Reply: Thanks for your comment and I have revised it which is shown below.

“Arsenic also plays an important role in the uniformity and endurance of selector devices, as shown in the current-voltage (I-V) curves of Ge-S-As devices **under** 100 continuous triangular pulses (**Fig. 1c**) and the performance statistics of individual devices (**Fig. S1**).”

2.

composition-independent switching speed is mainly due to the **pure electronic nature**
**of the OTS behavior**, in which atomic migration is barely involved^{25,26}.

Noe/Raty showed that the local atomic bond alignment changes the dielectric signature (Born charges) of the trap-related atoms under applied electric field. In that view, the word "pure" is not appropriate. Especially together with the word 'barely' in the next sentence. However, the 'barely' is indeed appropriate and one does not need/expect migration for OTS switching.

How about 'predominantly/mostly' instead of 'pure'?

Reply: Thanks for your comment and I really agree with your suggestions. The revised result is shown below

“A composition-independent switching speed is **predominantly** due to the electronic nature of the OTS behavior, in which atomic migration is barely involved^{19,20}.”

3.

process, in which the metal wire is bonded and the insulator layer is deposited^{28,29}. **Since**
**only chalcogenide glasses** exhibit threshold-switching behavior, if they transform into
a crystal under this condition, the crystallized selector then fails to exhibit any OTS

Highlight redacted 2023/4/12 16:41:24
Is that a proven fact? Do we have a reference?

Is that a proven fact? Do we have a reference?

Reply: Thanks for your comment. What we want to emphasize here is that OTS materials only show switching performance in the amorphous state, and they will fail after crystallization. We add the references and the revised result is shown below.

“Since **only OTS selectors in amorphous state** exhibit threshold-switching behavior⁸, if they transform into **crystals**, the crystallized selectors will lose the OTS function and can no longer be recovered.”

4.

of the OTS layer, before and after annealing. **e. Electrical performance for GeSAs₄₃**
**devices with different thicknesses, before and after heat treatment. V_{fire} , V_{th} , V_{h} and I_{on}**
**of both states increase while I_{off} monotonically decreases. **f. Annealed devices**
**demonstrate **more** superior endurance **p**roperties.**
Furthermore, we compare the performance variations of Ge-S-As devices with**

Sticky Note redacted 2023/4/18 14:20:43
Full/dashed lines are not clear what they stand for, please mention in the caption.

Full/dashed lines are not clear what they stand for, please mention in the caption.

Reply: Thanks for your comment and we add the captions which are shown below.

“**e. Electrical performance for as-deposited and annealed GeSAs₄₃ devices with different thicknesses. The solid and dashed lines represent the device performance before and after annealing, respectively. V_{fire} , V_{th} , V_{h} and I_{on} of both states increase while I_{off} monotonically decreases.**”

5.

demonstrate more superior endurance properties.

Furthermore, we compare the performance variations of Ge-S-As devices with different thicknesses of the functional layer, before and after annealing, as shown in Fig.

Does not sound right, it is like "more better", which is not correct.

Reply: Thanks for your comment. I have revised the descriptions according to comment 5 and 6. The result is shown below.

f. Endurances of annealed GeSAs devices. The lifetime of GeSAs₂₅ and GeSAs₄₃ devices is prolonged after annealing.

6.

demonstrate more superior endurance properties.

Furthermore, we compare the performance variations of Ge-S-As devices with

different thicknesses of the functional layer, before and after annealing, as shown in Fig.

"Superior endurance" is self-explanatory, whereas 'superior endurance properties' is not clear which properties.... Is it I_{on} , I_{off} or V_{th} (without drift) ?

Reply: Thanks for your comment. The device lifetime of annealed GeSAs₂₅ and GeSAs₄₃ devices is longer than that of as-deposited ones. The revised results are shown below.

f. Endurances of annealed GeSAs devices. The lifetime of GeSAs₂₅ and GeSAs₄₃ devices is prolonged after annealing.

7.

detail in Figs. S10-S13. In Fig. 2e, the solid lines represent the deposited devices, and

the dotted lines correspond to the annealed ones. Obviously, V_h is thickness-

independent, yet V_{fire} and V_{th} seem to increase nonlinearly as the thickness is doubled,

Here it is, maybe it belongs to the figure caption?

Reply: Thanks for your comment and we have moved this part to the caption. The revised results are shown below.

e. Electrical performance for as-deposited and annealed GeSAs₄₃ devices with different thicknesses. The solid and dashed lines represent the device performance before and after annealing, respectively. V_{fire} , V_{th} , V_h and I_{on} of both states increase while I_{off} monotonically decreases.

8.

222 2e (taking GeSAs₄₃ devices as an example). The changing tendencies are shown in

223 detail in Figs. S10-S13. In Fig. 2e, the solid lines represent the deposited devices, and

224 the dotted lines correspond to the annealed ones. Obviously, V_h is thickness-

on the fig S11c/d - 'before/'after' are missing the subject ('annealing')

Reply: Thanks for your comment. I have revised all the labels from **Figs.S11-S13** The revised results are shown below.

9.

Obviously, V_h is thickness-independent as the thickness is doubled, though I_{on} increases with thickness,

Sticky Note 2023/4/18 14:30:50

redacted 选项

Not that obvious for the reader. A reference would be helpful!

Not that obvious for the reader. A reference would be helpful. (V_h is thickness-independent)

Reply: Thanks for your comment. We have revised the related description and add the reference. The revised results are shown below.

[revised manuscript text omitted]

REVIEWER COMMENTS

Reviewer #1 (Remarks to the Author):

The authors provide a consistent review of the first manuscript version. Still some improvements can be done:

1) Even if the revised title is closer now to the real focus of the work, it would be better to make it even more clear, since it remains too generic. I suggest to replace “The Role of Arsenic in the Operation of Sulfur-based Electrical Threshold Switches” with “The role of Arsenic in the Operation of GeSAs based Electrical Threshold Switches”.

2) In the sentence “We added $x = 0, 20, 25$ and 43 at. % As into GeS, abbreviated as GeS, GeSAs20, GeSAs25 and GeSAs43, respectively” should be added clearly, what x represents (explained in methods but to be added in the beginning).

3) The picture quality is still too low. I recommend the use of vectorial images only. PNG, JPEG or similar low resolution formats should be avoided (except in case of HR)

4) In Fig. S7 the authors highlight a peak for GeS at 30° . I am still extremely skeptical about the GeS peak highlighted, and about its too much sharp shape. It is a minor peak, I would have expected the appearing of main ones before that. Moreover, the spread feature present in the acquired pattern certifies the still amorphous nature of the sample. Was the measurement performed at least twice? Could the peaks at about 34° be indexed as well, since they present a similar “intensity” as the one at 30° (close to noise)? Since the other samples were annealed up to 500°C , the same annealing would help on GeS sample to reinforce this hypothesis. Why it is not reported? In previous comment, the question was not about the pertinence of XRD technique to reveal diffraction peaks, but about the real possibility by XRD to highlight the structural changes. Fig. S8b on the contrary is reporting Raman spectra: it is more evident in this analysis that the GeS system has started an evolution wrt other samples. To me Raman results are more consistent with the authors’ hypothesis than XRD. Moreover, the reference used for XRD indexing is still not reported in the text.

5) R9-R14 results confirm the previous comment: ON current, as defined by the authors, is clearly a parameter induced by ON resistance plus the intrinsic switching voltage of the device. It is not a parameter of interest. Not clear why it is highlighted in Fig. S9 and how it is valuable, apart to highlight a possible different overshoot generated in the devices dependently on As content.

6) Sentences in the captions such as: “In 20 at. % As cells, V_{fire} is between 6 V and 7.69 V, V_{th} is in the range 2.42~4.41V, and V_{h} is located between 1.52 V and 2.12 V. In 25 at.% As cells, V_{fire} is between 5.73 V and 6.4 V, V_{th} is in the range 1.92~2.97 V, and V_{h} is located between 1.4 V and 1.7 V. In 43 at.% As cells, V_{fire} is between 4.79 V and 5.19 V, V_{th} is in the range 1.79~2.36 V, and V_{h} is located between 1.39 V and 1.57 V.” can be removed. They are not useful for the understanding, and the graph S10b is already there to summarize them. Same for other graphs.

“The on-current of the devices decreases from 0.71 mA to 0.4 mA, which is also consistent with the trend before annealing.”. ON current, as previously said, depends on ON resistance and voltage.

7) Line 140-150: no change has been done. As previously said, and confirmed by authors’ results ON current is not a parameter of interest for the devices comparison.

8) “Since I_{on} is almost size-independent, the current density of GeSAs devices sharply increases to >20 MA/cm² as the device size scales down to 60 nm, higher than Ge-Se/Te-based OTSs”. The sentence is not clear. Please consider revision.

9) “Although V_{th} is closely determined by the As content, the time spent in switching on and off seems to be As-independent”. Which is the “time spent in switching on and off”?

10) “The reason for this is due to using pulses is more relevant to the real application scenario than DC and the device cannot be repeated operating through DC operations because the width of the read pulse of DC test is 3 ms which may cause significant damage to the device.” This explanation can be added into the text.

11) If the RC can be estimated in the ten nanoseconds range, please add a resolution limit line/lines in Fig.S4d. Maybe the reference in the text to S4c for the method would clarify how the on/off time has been calculated.

12) Fig. R16b (not reported in the article and neither in supplementary material) contradicts the supposed benefit of As on drift. The sentence “In GeSAs43 devices, V_{th} drifts 700 mV after 1 day, which is consistent with the conclusion obtained in Fig. R16a, 25 at.% As incorporation can effectively suppress V_{th} drift even in cycled devices.” is not correct. Which is the link between GeSAs43 and 25at.% As incorporation? A proper calculation of the drift coefficient or a graph of the V_{th} variation along time would have certainly be more helpful than the reported trends. This sentence “For pure GeS, the V_{th} value of 1.46 V drifts to 1.62 V within 1 hour and then further rises to 1.72 V after 1 day. In the same way, the V_{th} value of GeSAs20 starts from 2.98 V, then increases to 3.24 V, and stops at 3.16 V. The rising trend of V_{th} was inhibited at 25 at. % As content, where V_{th} was around 2.6 V for all four tests, that is 2.64 V @ 0 s, 2.57 V @ 1 min, 2.42 V @ 1 hour and 2.75 V @ 1 day, i.e., only 0.15 V increasing, while V_{th} increases from 2.23 V to 2.54 V and further to 2.64 V @ 1 day in GeSAs43. Obviously, an appropriate As contents is sufficient to restrain the V_{th} drift.” is just confusing, and not useful for the understanding of the drift trend. “Obviously, an appropriate As content is sufficient to restrain the V_{th} drift” is not that obvious, or at least should be better explained based on data analyses. Is the V_{th} reduction for As 25at.% from 2.64V (variance? statistical error?) down to 2.42V after 1 hour understood? or the reduction for GeSAs20 from 3.24 V down to 3.16V? Error bars should help to reconsider the previous analysis.

13) R17 and R18 and Fig.1f can be better summarized: how many devices tested, an error bar to be used more than all the devices on same graph, etc. ? The trend is interesting, with a gradual degradation of the devices along cycling. A table to summarize the performances before and after annealing could help?

14) In the endurance plots, we can observe a clear evolution of the ON resistance of the device along cycling, and even if bottom right IV plots of S15 and S16 are not perfectly clear, they seem to reveal an

evolution of the material along cycling. S16 presents some problems in the legend? Which curve corresponds to which color?

The answer to the previous question was: "...we have updated all the figures. We really agree with what you said. The atomic migration/composition evolution were not excluded.". However, at line 307 and 308 the comment is unchanged. I did not find in the text the changes proposed. Fig. R19 is interesting, showing that in reality the segregation takes place at low cycles number and the channel is formed at the edge of the electrode (point effect). This is the region considered by the forming process. The rest of the layer is not considered, and then R21 to R23 are showing the result on a virgin layer, partially useful for the understanding. More valuable is to add R19 to the article, and same analysis could be extended to the other alloys at this point, since the trend observed is the same. The system is segregating under forming and following pulse applications. The composition/s of the initialized region should be the focus of the investigations, since likely responsible for the collected electrical results with its evolution along cycling. Is AsS system responsible in reality for the switching capability, and the fact to add more As, makes it more likely to form (despite segregation phenomena)?

15) "... we find that with the increase of As content, the fluctuation amplitude of I_{off} decreases with the increase of pulse counts which is shown in Fig. R17". I cannot appreciate such decrease. The author means that at 108 the off current increases for low As contents? In the annealed samples, it is more evident the degradation of the I_{off} dependently on As content.

16) Fig.4e: How we can consider 262, 255 and 248 cm^{-1} peaks attributed to completely different vibrations and not the same one shifting due to As introduction? The spectra of GeSAs systems are close, with same vibrational contributions that change in intensity.

17) As segregation, confirmed by R25, table R3 and Fig.6c (R26), can be better highlighted in the text.

Reviewer #2 (Remarks to the Author):

The manuscript is really well written and based on a thorough analysis of the data. Furthermore, all the issues raised by the different referees are addressed seriously and with a large amount of new measurements and supporting data. Now all the conclusions are well supported by the experimental results. Furthermore, measurements protocols and summary tables were provided to help the readers. Based on these considerations and the importance of the present work, I am of the opinion that the present manuscript is well suited for publication in Nature Communications and it can be published in the present form.

Reviewer #3 (Remarks to the Author):

The authors convinced me that the present manuscript is more than the previous publication with additional experimental measurements and insights on the role of the As. Additionally, the other concerns were addressed satisfactorily. As such, I recommend the manuscript for publication.

REVIEWER COMMENTS

Reviewer #1 (Remarks to the Author):

The authors provide a consistent review of the first manuscript version. Still some improvements can be done:

Dear Reviewer 1:

We are very grateful to Reviewer 1 for the interest in the theme of our manuscript. We've answered the questions point by point, which are as follows.

1) Even if the revised title is closer now to the real focus of the work, it would be better to make it even more clear, since it remains too generic. I suggest to replace “The Role of Arsenic in the Operation of Sulfur-based Electrical Threshold Switches” with “The role of Arsenic in the Operation of GeSAs based Electrical Threshold Switches”.

Reply: Many thanks for pointing out this issue. If the title is changed to GeSAs-based one, it seems too narrow to be published in Nature Communications, which requires wide readership. Thank you for your considerations.

2) In the sentence “We added $x = 0, 20, 25$ and 43 at. % As into GeS, abbreviated as GeS, GeSAs₂₀, GeSAs₂₅ and GeSAs₄₃, respectively” should be added clearly, what x represents (explained in methods but to be added in the beginning).

Reply: Thanks for your comments and I have revised it in the manuscript. The result of the modification in the sentence is shown below.

“We added $0, 20, 25$ and 43 at. % As into GeS, abbreviated as GeS, GeSAs₂₀, GeSAs₂₅ and GeSAs₄₃, respectively.”

3) The picture quality is still too low. I recommend the use of vectorial images only. PNG, JPEG or similar low-resolution formats should be avoided (except in case of HR)

Reply: Thanks for your comments. Indeed, the picture quality is still too low. In our manuscript, we inserted our images to make them easier to read for reviewers. In this submission, we would upload vectorial images.

4) In Fig. S7 the authors highlight a peak for GeS at 30° . I am still extremely skeptical about the GeS peak highlighted, and about its too much sharp shape. It is a minor peak, I would have expected the appearing of main ones before that. Moreover, the spread feature present in the acquired pattern certifies the still amorphous nature of the sample. Was the measurement performed at least twice? Could the peaks at about 34° be indexed as well, since they present a similar “intensity” as the one at 30° (close to noise)? Since the other samples were annealed up to 500°C , the same annealing would help on GeS sample to reinforce this hypothesis. Why it is not reported? In previous comment, the question was not about the pertinence of XRD technique to reveal diffraction peaks, but about the real possibility by XRD to highlight the structural changes. Fig. S8b on the contrary is reporting Raman spectra: it is more evident in this analysis that the GeS system has started an evolution wrt other samples. To me Raman results are more consistent with the authors' hypothesis than XRD. Moreover, the reference used for XRD indexing is still not reported in the text.

Reply: Thank you for your comment. To avoid the film evaporation during annealing, we usually deposit amorphous SiO₂ with a thickness of 5 nm on the surface of the film. The XRD results of GeS samples annealed at different temperatures are shown in **Fig. R1a**, where the peaks at 30° and 34° are consistent with the literature¹. When the annealing temperature exceeds 400 °C, it is necessary to increase the thickness of SiO₂ to 20 nm to ensure the existence of film after annealing. XRD patterns of GeS films treated at higher temperatures are shown in **Fig. R1b**. Compared to **Fig. R1a**, a new peak at 17° (<https://www.hqgraphene.com/GeS.php>) appeared in **Fig. R1b**, while the peaks at 30° and 34° exist, too. However, the surface of the film becomes wrinkled which may account for the differences between the GeS films annealed higher temperatures in **Fig. R1b**. In addition, **Fig. R1a** is sufficient to indicate that locally ordered structures have shown up in the GeS film after annealing at 400 °C, so we did not display XRD results at higher heat treatment temperatures. As for Raman results, we strongly agree with your view that Raman spectra more intuitively reflect the structural changes inside GeS after annealing than XRD results. The literature related to XRD has been added to the supplementary materials.

Fig. R1 a. XRD of GeS films covered with 5-nm SiO₂ annealed at different temperatures. **b.** XRD of GeS films covered with 20-nm SiO₂ annealed at higher temperatures

5) R9-R14 results confirm the previous comment: ON current, as defined by the authors, is clearly a parameter induced by ON resistance plus the intrinsic switching voltage of the device. It is not a parameter of interest. Not clear why it is highlighted in Fig. S9 and how it is valuable, apart to highlight a possible different overshoot generated in the devices dependently on As content.

Reply: Thanks for your comments. According to your statement, ON current is indeed not an intrinsic parameter. However, threshold voltage (V_{th}) which is the key parameter of OTS is also affected by factors such as the applied signal^{2,3} and the film thickness⁴. Whether a particular parameter of the device is investigated depends on the actual application requirements. In practical 3D PCM applications, the external resistance of OTS cannot be easily tailored, and the phase change material does require a current density more than 10 MA/cm² to melt⁵. Based on above facts, reducing the electrode size turns out to be a very effective way to increasing the current density which asks the device to maintain the performance during scaling down. In **Fig. S9**, all the devices containing As with 60-nm electrode operate normally after annealing which indicates the great potential of GeSAs in scalability. The emphasis on the ON currents of the different components of devices are intended to show that the devices with lower feature sizes can still be operated after device miniaturization with essentially no degradation in performance while providing a sufficient current density to drive the PCM at the same time.

6) Sentences in the captions such as: “In 20 at. % As cells, V_{fire} is between 6 V and 7.69 V, V_{th} is in the range 2.42~4.41 V, and V_{h} is located between 1.52 V and 2.12 V. In 25 at.% As cells, V_{fire} is between 5.73 V and 6.4 V, V_{th} is in the range 1.92~2.97 V, and V_{h} is located between 1.4 V and 1.7 V. In 43 at.% As cells, V_{fire} is between 4.79 V and 5.19 V, V_{th} is in the range 1.79~2.36 V, and V_{h} is located between 1.39 V and 1.57 V.” can be removed. They are not useful for the understanding, and the graph S10b is already there to summarize them. Same for other graphs.

Reply: Thank you for your comment. We deleted such sentences in the captions.

7) Line 140-150: no change has been done. As previously said, and confirmed by authors’ results ON current is not a parameter of interest for the devices comparison.

Reply: Thank you for your comment. As indicated in the answer to Q5, ON current plays an integral role in characterizing the drive capability of the OTS device, although this parameter is affected by the resistance value of the external resistor. In addition, other parameters of the device have been described in other parts of the paper, and the description of I_{on} is not redundant.

8) “Since I_{on} is almost size-independent, the current density of GeSAs devices sharply increases to $>20 \text{ MA/cm}^2$ as the device size scales down to 60 nm, higher than Ge-Se/Te-based OTSs”. The sentence is not clear. Please consider revision.

Reply: Thank you for your comment. This statement is directly related to the formula for calculating the ON current density (J_{on}) ($J_{\text{on}} = \frac{I_{\text{on}}}{\pi r^2}$, r refers to the radius of the cylindrical electrode). As I_{on} is almost unaffected by device miniaturization, while a reduction in electrode size will lead to a significant increase of J_{on} . We originally wanted to express that under the same device structure and feature size, GeSAs devices has an advantage in drive currents. But the original expression may be not clear enough, so it seems to lack a certain coherence. The result of the modification in the sentence is shown below.

“Since I_{on} is almost size-independent, the current density of GeSAs devices sharply increases to $>20 \text{ MA/cm}^2$ as the device size scales down to 60 nm, higher than that of Ge-Se/Te-based OTSs with the same size.”

9) “Although V_{th} is closely determined by the As content, the time spent in switching on and off seems to be As-independent”. Which is the “time spent in switching on and off”?

Reply: Thank you for your comment. “time spent in switching on and off” means literally what it says, that is, the time it takes to turn on and off, often referred as switching speed as shown in Fig. S4c. This expression seems easy to mislead the readers, we have corrected it. The result of the modification in the sentence is shown below.

“Although V_{th} is closely determined by the As content, the switching speed seems to be As-independent.”

10) “The reason for this is due to using pulses is more relevant to the real application scenario than DC and the device cannot be repeated operating through DC operations because the width of the read pulse of DC test is 3 ms which may cause significant damage to the device.” This explanation can be added into the text.

Reply: Thank you for your suggestions, the explanation does help readers to more intuitively

understand our experimental methods and limitations, and we have added specific descriptions in Methods.

“The device performances were characterized by applying pulses through a Keithley 4200A-SCS instrument rather than DC test because of the significant damage to the device caused by DC test.”

11) If the RC can be estimated in the ten nanoseconds range, please add a resolution limit line/lines in Fig.S4d. Maybe the reference in the text to S4c for the method would clarify how the on/off time has been calculated.

Reply: Thanks for your comment. We have minimized the effect of RC under the allowed experimental conditions. The smallest time interval that the oscilloscope can recognize is 0.1 ns, which is determined by the bandwidth of the oscilloscope. However, the practical problem in the speed test is that our measurement method itself will affect the parameters of the device. We have added grid lines to **Fig. S4c** to make it easier for you to understand.

Fig. R2 On & off speed of GeSAs device

12) Fig. R16b (not reported in the article and neither in supplementary material) contradicts the supposed benefit of As on drift. The sentence “In GeSAs₄₃ devices, V_{th} drifts 700 mV after 1 day, which is consistent with the conclusion obtained in Fig. R16a, 25 at.% As incorporation can effectively suppress V_{th} drift even in cycled devices.” is not correct. Which is the link between GeSAs₄₃ and 25at.% As incorporation?

A proper calculation of the drift coefficient or a graph of the V_{th} variation along time would have certainly be more helpful than the reported trends.

This sentence “For pure GeS, the V_{th} value of 1.46 V drifts to 1.62 V within 1 hour and then further rises to 1.72 V after 1 day. In the same way, the V_{th} value of GeSAs₂₀ starts from 2.98 V, then increases to 3.24 V, and stops at 3.16 V. The rising trend of V_{th} was inhibited at 25 at. % As content, where V_{th} was around 2.6 V for all four tests, that is 2.64 V @ 0 s, 2.57 V @ 1 min, 2.42 V @ 1 hour and 2.75 V @ 1 day, i.e., only 0.15 V increasing, while V_{th} increases from 2.23 V to 2.54 V and further to 2.64 V @ 1 day in GeSAs₄₃. Obviously, an appropriate As contents is sufficient to restrain the V_{th} drift.” is just confusing, and not useful for the understanding of the drift trend.

“Obviously, an appropriate As content is sufficient to restrain the V_{th} drift” is not that obvious, or at least should be better explained based on data analyses. Is the V_{th} reduction for As 25at.% from

2.64V (variance? statistical error?) down to 2.42V after 1 hour understood? or the reduction for GeSAs₂₀ from 3.24 V down to 3.16V? Error bars should help to reconsider the previous analysis.

Reply: Thank you for your comments. After 1000 operations, the V_{th} of GeSAs₂₀ device drifts 460 mV, the V_{th} of GeSAs₂₅ device increases by 440 mV, and the GeSAs₄₃ device has a change in V_{th} of 700 mV after 1 day. But the V_{th} of GeS device drifts only 255 mV. Therefore, the V_{th} fluctuation of GeS device is the smallest under this situation, which may be due to phase separation that causes significant fluctuations in device performance. **Fig. R3a** has been added in the manuscript and related descriptions has been modified as follows:

“For pure GeS, the average V_{th} increases by 196 mV within 1 min and then further rises up another 68 mV after 1 day. In the same way, the variation value of V_{th} of GeSAs₂₀ starts from 106 mV, then increases to 265 mV, and stops at 182 mV. The rising trend of V_{th} was inhibited at 25 at. % As content, where the variation value of V_{th} decreases by 65 mV @ 1 min and eventually goes up 113 mV @ 1 day, while the increase in V_{th} climbs up from 314 mV to 416 mV @ 1 day in GeSAs₄₃.”

Fig. R3 a. V_{th} drift of as-deposited GeSAs devices with time. **b.** V_{th} drift of GeSAs devices after 1000-time operations.

13) R17 and R18 and Fig.1f can be better summarized: how many devices tested, an error bar to be used more than all the devices on same graph, etc.? The trend is interesting, with a gradual degradation of the devices along cycling. A table to summarize the performances before and after annealing could help?

Reply: Thank you for your comments and we have summarized the endurance of GeSAs devices as shown in **Fig. R4**. In **Fig. R4a**, 1 GeS, 2 GeSAs₂₀, 2 GeSAs₂₅ and 5 GeSAs₄₃ as-deposited devices are tested. Since DC testing causes a destructive damage to the device, it leads to larger differences in current values of different devices under the same amount of rectangle pulses. It also affects the yield in endurance testing. Similarly, 2 GeSAs₂₀, 5 GeSAs₂₅ and 3 GeSAs₄₃ annealed devices are tested which are shown in **Fig. R4b**. That is the reason why we do not choose to present the endurance performance using error bars. A comparison of the device properties before and after annealing is shown in **Table. R1**.

Fig. R4 a. Endurance summary of GeSAs devices. b. Endurance summary of annealed GeSAs devices.

Summary of GeSAs device performances before and after annealing using different materials.									
Material	Feature Size (nm)	Thickness (nm)	Selectivity	I_{OFF} (nA)	V_{th} (V)	V_h (V)	Speed (ns)	Endurance	
Before annealing	GeSAs ₂₀	200	10	10^5	11	2.8	1.5	~10	10^8
	GeSAs ₂₅	200	10	10^5	13	2.6	1.5	~10	10^8
	GeSAs ₄₃	200	10	10^5	20	2.2	1.4	~10	10^8
After annealing	GeSAs ₂₀	200	10	10^5	2.3	3.3	1.9	~10	10^7
	GeSAs ₂₅	200	10	10^5	12	2.5	1.5	~10	10^9
	GeSAs ₄₃	200	10	10^4	15	2	1.4	~10	9×10^9

Table. R1 Summary of GeSAs device performances before and after annealing using different materials.

14) In the endurance plots, we can observe a clear evolution of the ON resistance of the device along cycling, and even if bottom right IV plots of S15 and S16 are not perfectly clear, they seem to reveal an evolution of the material along cycling. S16 presents some problems in the legend? Which curve corresponds to which color?

Reply: Thank you for your comments. We have updated the DC test plot in **Fig. S16** as shown below.

Fig. R5 I_{off} of annealed GeSAs₄₃ device under every endurance tests

The answer to the previous question was: "...we have updated all the figures. We really agree with what you said. The atomic migration/composition evolution were not excluded.". However, at line 307 and 308 the comment is unchanged. I did not find in the text the changes proposed. Fig. R19 is

interesting, showing that in reality the segregation takes place at low cycles number and the channel is formed at the edge of the electrode (point effect). This is the region considered by the forming process. The rest of the layer is not considered, and then R21 to R23 are showing the result on a virgin layer, partially useful for the understanding. More valuable is to add R19 to the article, and same analysis could be extended to the other alloys at this point, since the trend observed is the same. The system is segregating under forming and following pulse applications. The composition/s of the initialized region should be the focus of the investigations, since likely responsible for the collected electrical results with its evolution along cycling. Is AsS system responsible in reality for the switching capability, and the fact to add more As, makes it more likely to form (despite segregation phenomena)?

Reply: Thank you for your comments. We did add the relevant description as shown below.

“The I_{on} is very stable, whereas the I_{off} fluctuates with an apparent upward drift. We assume that the increase in I_{off} may be jointly caused by phase separation or the formation of Ge-Ge filaments.”

Indeed, the reason for the evolution of I_{off} during the cycling operations are an interesting topic. To do the TEM investigation, TEM lamella needed to be prepared first. We need to do many times to get a TEM lamella from the devices with 60-200 nm electrode. It really time-consuming process for investigating four devices. Thanks very much for your consideration.

15) “... we find that with the increase of As content, the fluctuation amplitude of I_{off} decreases with the increase of pulse counts which is shown in Fig. R17”. I cannot appreciate such decrease. The author means that at 10^8 the off current increases for low As contents? In the annealed samples, it is more evident the degradation of the I_{off} dependently on As content.

Reply: Thank you for your comment. In **Fig. R17**, with the increase of As content, as the As content increases, I_{off} exhibits better stability in endurance tests. This is not just comparing the I_{off} values at 10^8 cycles, but rather stating that as the number of testing pulses increases, the I_{off} fluctuations of As-rich devices are relatively small, which is very evident in the **Fig. R17**. In the annealed sample test of **Fig. R18**, the same conclusion is still reached. It is obvious that the endurance of 20 at.% As device only reaches 10^7 , GeSAs₂₅ device reaches 10^9 , while GeSAs₄₃ device can reach $\sim 10^{10}$ whose overall I_{off} fluctuation is small. The two conclusions are consistent and there is no conflict.

16) Fig.4e: How we can consider 262, 255 and 248 cm^{-1} peaks attributed to completely different vibrations and not the same one shifting due to As introduction? The spectra of GeSAs systems are close, with same vibrational contributions that change in intensity.

Reply: Thank you for your comment. The peak positions in Raman spectroscopy correspond to the frequency of coupling vibrations between atoms in the material. In other words, the peak position of dominant atomic group is basically unchanged. In GeSAs₂₀, the concentration of Ge and S atoms is relatively high. When the As content increases to 25 at.%, the peak position remains relatively unchanged compared to GeSAs₂₀ due to the small concentration changes of Ge, S and As atoms. When the As content reaches 43 at.%, the S element content decreases significantly and the As content increases significantly which corresponds to the appearance of As-As cluster.

17) As segregation, confirmed by R25, table R3 and Fig.6c (R26), can be better highlighted in the text.

Reply: Thank you for your comment and we have added the relevant description of the As segregation. The results are shown below.

“Interestingly, the contribution of Ge-Ge chains to the trap states appears to become saturated when the As content reaches 20 at. %, thereafter the total density of in-gap states starts to decrease, even though the As-As bonds provide extra traps (**Fig. 5**) which results from As separation.”

In particular, new trap states are found in GeSAs₄₃, mainly because the excess As atoms can induce As segregation, in which homopolar As-As bonds generate extra free electrons.

Reference

1. Lan, C., Li, C., Yin, Y., Guo, H. & Wang, S. Synthesis of single-crystalline GeS nanoribbons for high sensitivity visible-light photodetectors. *J. Mater. Chem. C* **3**, 8074–8079 (2015).
2. Yoo, J. *et al.* Threshold Voltage Drift in Te-Based Ovonic Threshold Switch Devices Under Various Operation Conditions. *IEEE Electron Device Lett.* **41**, 191–194 (2020).
3. Chai, Z. *et al.* Dependence of Switching Probability on Operation Conditions in Ge_xSe_{1-x} Ovonic Threshold Switching Selectors. *IEEE Electron Device Lett.* **40**, 1269–1272 (2019).
4. Jia, S. *et al.* Ultrahigh drive current and large selectivity in GeS selector. *Nat. Commun.* **11**, 4636 (2020).
5. Shen, J. *et al.* Elemental electrical switch enabling phase segregation-free operation. *Science* (80-.). **374**, 1390–1394 (2021).

REVIEWERS' COMMENTS

Reviewer #1 (Remarks to the Author):

Thanks to the authors for the feedback and answers. Below my final comment, so I leave it to the editor to consider the article for publication based on his feedback.

Comment 13: Please report the devices number considered for the test of endurance in the text.

REVIEWER COMMENTS

Reviewer #1 (Remarks to the Author):

Thanks to the authors for the feedback and answers. Below my final comment, so I leave it to the editor to consider the article for publication based on his feedback.

Comment 13: Please report the devices number considered for the test of endurance in the text.

Reply: Thank the reviewers for recognizing our answers. About ten devices were used in the test of endurance.